# Contributions of associative and non-associative learning to the dynamics of defensive ethograms

**Quan-Son Eric Le[1,2], Daniel Hereford[1,2], Chandrashekhar D Borkar[1,3], Zach Aldaco[1,3], Julia Klar[1,2], Alexis Resendez[1,3], Jonathan P Fadok[1,3]***

[1]Tulane Brain Institute, Tulane University, New Orleans, United States; [2]Program in Neuroscience, Tulane University, New Orleans, United States; [3]Department of Psychology, Tulane University, New Orleans, United States

## eLife Assessment

This study is deemed to be an **important** work that carefully deconstructs multi-faceted conditioned fear behavior in mice. The well-controlled experiments provide **convincing** data that will be of interest to other researchers in the field.

*For correspondence:
jfadok@tulane.edu

Competing interest: The authors declare that no competing interests exist.

**Abstract** Defensive behavior changes based on threat intensity, proximity, and context of exposure, and learning about danger-predicting stimuli is critical for survival. However, most Pavlovian fear conditioning paradigms focus only on freezing behavior, obscuring the contributions of associative and non-associative mechanisms to dynamic defensive responses. To thoroughly investigate defensive ethograms, we subjected male and female adult C57BL/6 J mice to a Pavlovian conditioning paradigm that paired footshock with a serial compound stimulus (SCS) consisting of distinct tone and white noise (WN) stimulus periods. To investigate how associative and non-associative mechanisms affect defensive responses, we compared this paired SCS-footshock group with four control groups that were conditioned with either pseudorandom unpaired presentations of SCS and footshock, shock only, or reversed SCS presentations with inverted tone-WN order, with paired or unpaired presentations. On day 2 of conditioning, the paired group exhibited robust freezing during the tone period with switching to explosive jumping and darting behaviors during the WN period. Comparatively, the unpaired and both reverse SCS groups expressed less tone-induced freezing and rarely showed jumping or darting during WN. Following the second day of conditioning, we observed how defensive behavior changed over two extinction sessions. During extinction, the tone-induced freezing decreased in the paired group, and mice rapidly shifted from escape jumping during WN to a combination of freezing and darting. The unpaired, unpaired reverse, and shock-only groups displayed defensive tail rattling and darting during the SCS, with minimal freezing and jumping. Interestingly, the paired reverse group did not jump to WN, and tone-evoked freezing was resistant to extinction. These findings demonstrate that non-associative factors promote some defensive responsiveness, but associative factors are required for robust cue-induced freezing and high-intensity flight expression.

## Introduction

Defensive responses have evolved to maximize survival (*Anderson and Adolphs, 2014*), and animals rapidly switch behaviors depending on threat imminence, the context of exposure, and previous experience with stimuli (*Perusini and Fanselow, 2015*). Understanding the mechanisms underlying

**eLife digest** Post-traumatic stress disorder (or PTSD for short) is a condition that can cause people to overreact to harmless cues, vividly re-experience a traumatic event, or freeze in place. To understand why this happens, researchers often study fear responses using an approach called fear conditioning, where laboratory animals learn to associate the sound of a tone with a mild electric shock. This conditioning causes animals to freeze with fear when they hear the tone.

However, focusing on freezing overlooks the range of defensive actions animals may carry out, such as escaping or fighting. Capturing this complexity in experiments is important for understanding the dynamic nature of fear responses that occur in PTSD. Previous work showed that conditioning mice with a two-part cue, such as a tone followed by white noise, caused mice to freeze during the first cue and jump during the second cue. However, whether the mice learned this behaviour through conditioning or if it was an instinctive response to the cues remained unclear.

To investigate this phenomenon, Le et al. – including some of the researchers involved in the previous work – conditioned mice with a variety of different cue combinations and monitored how they responded. As before, mice conditioned to associate a tone followed by white noise with an electric shock froze when they heard the tone and transitioned to jumping during the white noise. However, if during conditioning the sounds and shocks occurred at unpredictable times, the mice did not associate the sounds with the shock and therefore they froze less and rarely jumped. Similarly, reversing the order of the sounds so that the white noise happened before the tone also reduced jumping but not freezing.

To investigate whether the mice could unlearn this fear response, Le et al. exposed the fear-conditioned mice to the cues without an accompanying electric shock. The mice that had been conditioned with a tone followed by white noise showed a weaker response to the cues, only freezing and not jumping. However, the mice with the reversed cues still froze even after this exposure, and the mice with the non-associated cues maintained very little freezing and jumping.

Taken together, the findings suggest that while fear responses can be influenced by the association between certain noises and an electric shock, other factors such as the timing and the order of the sound cues can also impact the intensity of the fear response. The experiments also showed that this method of fear conditioning can be used for both learning and unlearning fear responses, revealing an approach for future studies into how fear responses change over time. Combining this more complex approach with other experimental techniques could help researchers identify the brain regions that drive fear responses, which may eventually benefit people with PTSD and other fear disorders.

adaptive defensive behavior may grant insight into the pathophysiology of post-traumatic stress and panic disorders, wherein heightened responses to external stimuli are observed, yet neuroscientists need more tractable methods with which to investigate how the nervous system controls complex experience-dependent behavior.

Pavlovian fear conditioning has been widely used as a model system to understand the neural mechanisms underlying fear-related learning and memory (*Bolles, 1970*; *Bolles and Collier, 1976*; *Grewe et al., 2017*; *Roy et al., 2017*; *Bouton et al., 2021*). In standard Pavlovian conditioning paradigms, freezing is the dominant defensive behavior evoked by contexts and learned cues that are paired with an aversive unconditioned stimulus (US), like footshock (*Blanchard and Blanchard, 1969*; *Bolles and Collier, 1976*). Other defensive responses like escape jumping (*Chu et al., 2024*) and darting (*Gruene et al., 2015*) are measured less often within conditioning, limiting insight into defensive response dynamics. To address this critical need, we developed a modified Pavlovian conditioning paradigm that elicits both freezing and flight behaviors in response to conditioned stimuli (*Fadok et al., 2017*; *Borkar et al., 2020*; *Borkar and Fadok, 2021*; *Borkar et al., 2024*), findings that have been replicated by others in both mice and rats (*Dong et al., 2019*; *Totty et al., 2021*). In this paradigm, mice are conditioned with a SCS consisting of a pure tone followed by WN, which terminates with a strong electrical footshock. After conditioning, mice exhibit contextual freezing which significantly increases in response to tone, and mice switch to robust flight responses upon WN presentation. These findings demonstrate that the magnitude and mode of defensive behavior

change with the psychological distance of threat, consistent with the predatory imminence continuum theory (*Perusini and Fanselow, 2015*).

However, the influence of non-associative elements on this ethological profile has recently been discussed (*Fanselow et al., 2019*; *Hersman et al., 2020*; *Trott et al., 2022*). It has been suggested that the inherent salience of the WN stimulus contributes more to WN-evoked flight response than its predictive association with the US (*Hersman et al., 2020*). Others claim that the immediate transition from freezing to flight behavior is a result of the rapid change and relative increase in stimulus intensity from tone to WN that is caused by non-associative sensitization, or by inherent stimulus properties, akin to an acoustic startle response (*Fanselow et al., 2019*; *Trott et al., 2022*). In addition, sensitization and stimulus salience are known to intensify freezing responses to auditory stimuli (*Kamprath and Wotjak, 2004*), and mice show increased reactivity to a WN stimulus after experiencing stress (*Hoffman et al., 2022*). These findings highlight the need to better elucidate the associative and non-associative elements of Pavlovian fear conditioning that influence the expression of defensive behavior.

To address this, we utilized four control groups for the non-associative effects of conditioning. To test the importance of the SCS-shock contingency, we utilized an unpaired control procedure in which the US and the SCS were presented in a separated, pseudorandom, and non-predictive fashion (*Rescorla, 1967*). To test the effects of sensitization by the shock, we presented footshock alone during conditioning. To test the impact of stimulus intensity and salience, we conducted paired and unpaired conditioning using a reversed SCS where the WN preceded the tone. We compared these four control groups against a paired SCS-shock conditioning group to determine the effects of associative learning on SCS-evoked fear behavior. All groups went through two extinction sessions with SCS presentations alone to elucidate the extent to which prior associative pairing affects de-escalating response strategies, as well as to identify defensive behaviors that are expressed in the absence of a strict threat-signaling association.

## Results

### Stimulus-evoked freezing and activity are affected by SCS-shock contingency and stimulus order

Equal numbers of male and female mice were randomly assigned to either a paired (**PA**), unpaired (**UN**), shock-only (**SO**), paired reverse (**PA-R**), or unpaired reverse (**UN-R**) group for fear conditioning and fear extinction training (*Figure 1*). Data from the PA, UN, and SO groups were statistically tested for sex differences and the significant results from these analyses are listed in *Table 1*. Given that most comparisons did not yield significant differences, data from male and female mice were pooled for statistical comparisons between groups. Additionally, given the minimal sex differences observed within the PA, UN, and SO groups, we reduced the number of subjects in the PA-R and UN-R groups and, therefore, did not statistically test for sex differences.

Behavioral data from the PA, UN, PA-R, and UN-R groups during the second day of fear conditioning (CD2) were compared to observe how conditioned defensive behavior differs based on the associative value and stimulus order of the SCS. A two-way ANOVA was used to analyze the effect of trial and group on freezing during the tone and WN. There was no statistically significant interaction between the trial and group for tone-induced freezing (*Figure 2A*; $F_{(12, 340)}=0.65$, p=0.80); however, there was a significant main effect of the trial ($F_{(4, 340)}=6.3$, p<0.0001) and group ($F_{(3, 340)}=19.6$, p<0.0001). All groups showed little freezing to the WN (*Figure 2B*), and no significant interaction between trial and group ($F_{(12, 340)}=0.76$, p=0.69) or main effect of the trial ($F_{(4, 340)}=2.07$, p=0.08) were found. We did find a main effect of group ($F_{(3, 340)}=3.00$, p=0.03), which was attributed to greater freezing from the PA-R group during Trial 1.

An activity index was calculated for each mouse as a combined measurement of cue-induced locomotion with escape jumping (see Methods), and a two-way ANOVA was used to analyze tone- and WN-evoked activity indices (*Figure 2C and D*). The activity indices for all groups were very low during the tone period (*Figure 2C*), and there was no significant interaction between the trial and group ($F_{(12, 340)}=0.98$, p=0.47). The activity index during the tone decreased over trials, concomitant with the observed increase in freezing behavior (main effect of trial, $F_{(4, 340)}=3.58$, p=0.007). There was a significant effect of group ($F_{(3, 340)}=2.7$, p=0.045), which was attributed to the UN-R group displaying higher

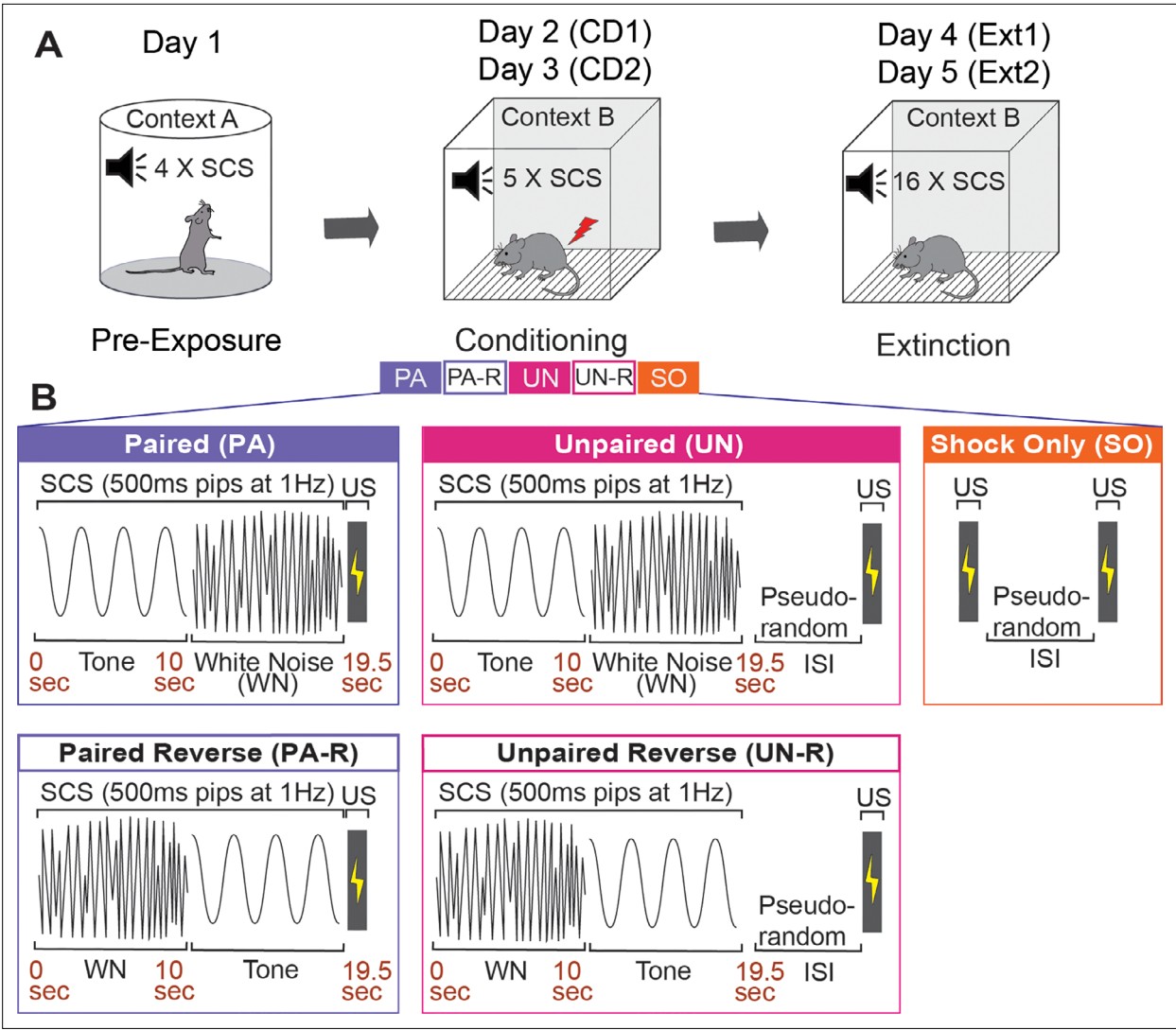

**Figure 1.** Experimental design. (**A**) Graphical representation of the three stages of the SCS conditioning paradigm. (**B**) Five SCS-shock association variants were used during conditioning. SCS, Serial compound stimulus; CD1, Conditioning Day 1; CD2, Conditioning Day 2; Ext1, Extinction Day 1; Ext2, Extinction Day 2; US, Unconditioned stimulus; ISI, Inter-stimulus interval.

activity during Trial 2. While the WN-evoked activity indices in all groups showed no significant trial by group interaction ($F_{(12, 340)}=0.15$, $p=0.99$), or main effect of the trial ($F_{(4, 340)}=0.54$, $p=0.70$), a significant main effect of group (*Figure 2D*, $F_{(3, 340)}=9.03$, $p<0.0001$) was observed.

An ordinary one-way ANOVA was used to compare average freezing and activity indices between all groups, and Tukey's multiple comparisons test was used for post-hoc comparisons. The PA and PA-R group showed significantly higher freezing during the tone than the UN group (*Figure 2E*, $F_{(3, 68)}=9.56$, $p<0.0001$; PA vs UN, $p<0.0001$; PA-R vs UN, $p=0.046$), and there was no significant difference between groups in their activity indices during tone (*Figure 2F*, $F_{(3, 68)}=2.14$, $p=0.10$). On the contrary, while there were no differences in WN-evoked freezing between groups (*Figure 2G*, $F_{(3, 68)}=2.62$,

**Table 1.** Statistical analysis of sex differences in defensive behavior.

| Figure | Group | Behavioral Comparison | Statistical Test | p-value | Result |
|---|---|---|---|---|---|
| 2E | UN | Average tone-evoked % freezing | Welch's unpaired t-test | p=0.0053 | Males froze more to tone than females in CD2 |
| 2F | UN | Average tone-evoked activity index score | Welch's unpaired t-test | p=0.0178 | Males had higher activity to tone compared to females in CD2 |

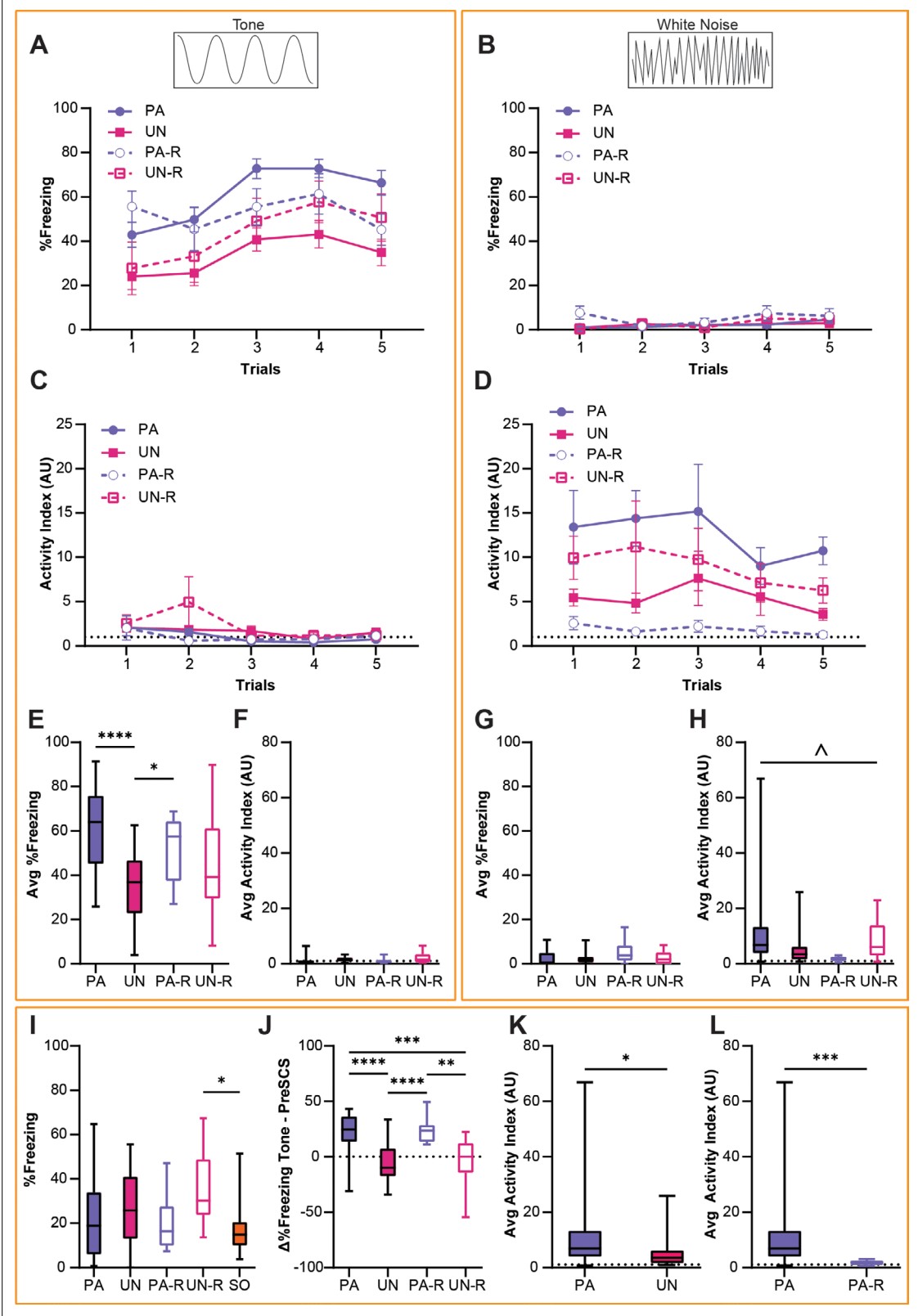

**Figure 2.** Stimulus-evoked freezing and activity during CD2 are affected by serial compound stimulus (SCS)-shock contingency. (**A**) Trial-by-trial freezing during the tone period. (**B**) Trial-by-trial freezing during the white noise (WN) period. (**C**) Trial-by-trial activity index during the tone period. (**D**) Trial-by-trial activity index during the WN period. (**E**) Average freezing during the tone period from all trials of CD2. (**F**) Average activity index scores during the tone period from all trials of CD2. (**G**) Average freezing during the WN period from all trials of CD2. (**H**) Average activity index scores during the

*Figure 2 continued on next page*

*Figure 2 continued*

WN period from all trials of CD2. (**I**) Baseline contextual freezing levels during CD2. (**J**) Differences in freezing between pre-SCS and tone periods from all trials of CD2. (**K**) Average activity index scores during the WN period for the PA and UN groups from all trials of CD2. (**L**) Average activity index scores during the WN for the PA and PA-R groups from all trials of CD2. PA: n=32, UN: n=20, PA-R: n=10, UN-R: n=10, SO: n=20. Data from (**A–D**) are presented as mean ± SEM and were analyzed with two-way ANOVA. Data from (**E–J**) are presented as box-and-whisker plots from min to max and were analyzed with one-way ANOVA and Tukey's post-hoc multiple comparisons test. Data from (**K and L**) are presented as box-and-whisker plots from min to max and were analyzed with Welch's unpaired t-test. *p<0.05; **p<0.01; ***p<0.001; ****p<0.0001; ^p<0.05, effect of group. For both the standard and reverse SCS groups, (**A, C, E and F**) detail responses during the tone period, and (**B, D, G and H**) detail responses during the WN period.

The online version of this article includes the following source data for figure 2:

**Source data 1.** Dataset contains behavioral measurements for *Figure 2* from each individual mouse in each group, classified by sex, group, and trial where applicable.

p=0.058), there was a significant interaction when analyzing activity indices during WN (*Figure 2H*, $F_{(3, 68)}$=3.04, p=0.035). However, post-hoc comparisons did not yield significant pairwise differences.

An ordinary one-way ANOVA was used to test for between-group differences in contextual freezing during the initial 3 min of the session preceding the first SCS presentation, and Tukey's multiple comparisons test was used for post-hoc comparisons. There was a main effect of group for baseline contextual freezing (*Figure 2I*, $F_{(3, 68)}$=3.18, p=0.017), but significant differences were found only between the UN-R and SO groups (UN-R vs SO, p=0.013). Welch's unpaired t-test was used to compare the differences in freezing during the pre-SCS and tone periods to determine the extent to which the tone increased freezing. The PA and PA-R groups showed significantly greater increases in freezing from pre-SCS to tone compared to the UN and UN-R groups (*Figure 2J*, $F_{(3, 68)}$=15.92, p<0.0001; PA vs UN, p<0.0001; PA vs UN-R, p=0.0009; PA-R vs UN, p<0.0001; PA-R vs UN-R, p=0.0055).

Given the interaction found in *Figure 2H*, we performed Welch's unpaired t-test to conduct pairwise comparisons between average activity indices during WN. WN-evoked activity was higher in the PA group compared to the UN group (*Figure 2K*, PA vs UN, $t_{(43.08)}$=2.36, p=0.023) and the PA-R group (*Figure 2L*, PA vs PA-R, $t_{(31.62)}$=3.89, p=0.0005). Additionally, the PA-R group displayed lower WN-evoked activity than both of the UN groups (PA-R vs UN, $t_{(20.65)}$=2.65, p=0.015; PA-R mean ± SEM = 1.86±0.28; UN mean ± SEM = 5.39±1.31) and the UN-R group (PA-R vs UN-R, $t_{(9.215)}$=2.75, p=0.022; UN-R mean ± SEM = 8.84±2.52). There was no difference between the UN and UN-R groups with regards to WN-evoked activity (UN vs UN-R, $t_{(14)}$ = 1.214, p=0.24). While the PA and UN groups significantly differed in average activity indices during WN, the PA and UN-R groups did not (PA vs UN-R, $t_{(30.42)}$=0.9962, p=0.33; PA mean ± SEM = 12.54±2.73).

Overall, these data show that the respective changes in defensive behavior during tone and WN were significantly affected by the explicit pairing of SCS and shock during fear conditioning, and that the order of tone and WN presentation influenced the intensity of WN-evoked responses.

## Associative pairing of the SCS and shock elicits escape jumping and darting responses to WN after conditioning

Although the UN or UN-R groups did not receive an associative pairing between SCS and shock like the PA and PA-R groups, all groups still displayed increased activity indices to WN (*Figure 2D*). To determine if this behavioral response was due to defensive flight or a more basic locomotor response, we investigated the occurrence of escape jumping and darting behaviors during the WN presentation on CD2. A substantial percentage of PA mice jumped during WN on every trial, and these jumps were distributed across the entire WN period (*Figure 3A and B*). In contrast, an exceedingly small percentage of the UN group jumped during WN (*Figure 3C*), and when jumps occurred, they occurred at the onset of the WN (*Figure 3D*). Both groups responded to shock with a similar number of jumps (*Figure 3B and D*). Trial-by-trial, PA mice displayed more jumping behavior across the WN period (*Figure 3E*) compared to UN mice (*Figure 3F*). Like the UN group, the reverse SCS groups also displayed lower jumping percentages to WN, with jumps in the PA-R group occurring rarely near the transition from WN to tone, (*Figure 3G*) and jumps in the UN-R group occurring near WN onset (*Figure 3H*). The PA group had the largest percentage of mice that jumped to WN during CD2 compared to all other groups (*Figure 3I*). Lastly, all groups exhibited jumps to shock (*Figure 3J*), with greater percentages of the unpaired and reverse cohorts responding with jumps.

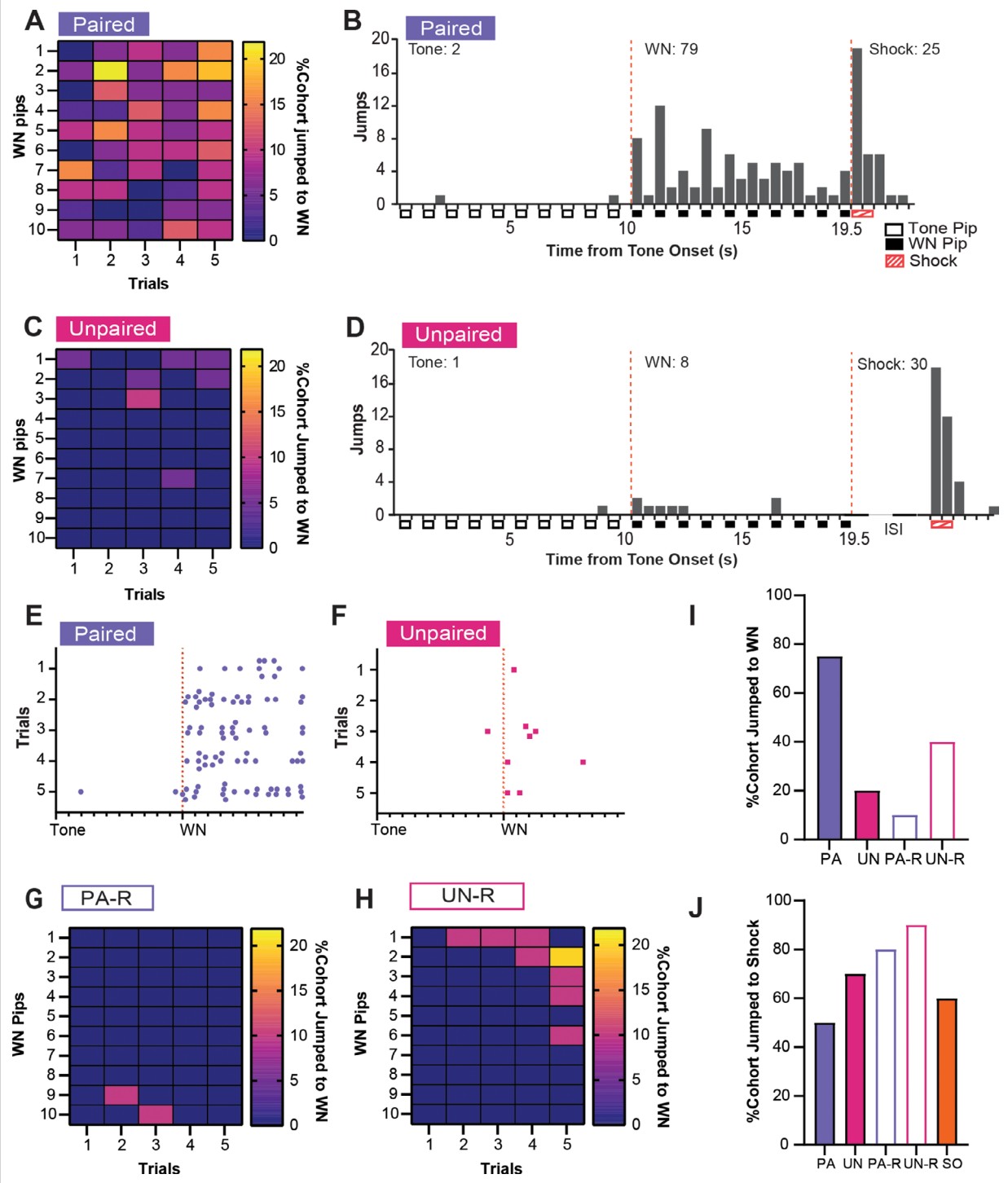

**Figure 3.** Associative pairings of the serial compound stimulus (SCS) and shock lead to robust escape jumping during white noise (WN). (**A**) The percentage of the paired (PA) group that exhibited jumping during WN on CD2. Data are distributed across 1 s bins, each coinciding with one of the ten pips of WN that occurred during each SCS presentation. (**B**) The cumulative distribution of jumps from 20 randomly selected subjects of the PA group across the duration of the SCS from all five trials of CD2. Empty boxes represent each 0.5 s pip of tone, filled boxes represent each 0.5 s pip of WN, and striped boxes represent the 1 s shock stimulus. The vertical dotted lines depict the onset and termination of the WN period. Total jumps per stimulus are listed above the histogram bars. (**C**) The percentage of the UN group that exhibited jumping during WN on CD2. Data are distributed across 1 s bins, each coinciding with one of the ten pips of WN that occurred during each SCS presentation. (**D**) The cumulative distribution of jumps from the UN group across the duration of the SCS from all five trials of CD2. Empty boxes represent each 0.5 s pip of tone, filled boxes represent each 0.5 s pip of WN, and striped boxes represent the 1 s shock stimulus. The vertical dotted lines depict the onset and termination of the WN period. ISI represents

*Figure 3 continued on next page*

*Figure 3 continued*

the period between SCS and shock. Total jumps per stimulus are listed above the histogram bars. € The distribution of jumps across the duration of the SCS from 20 randomly selected subjects of the PA group for each trial of CD2. Each dot represents a single jump event, and each tick on the x-axis represents the onset of each pip of tone or WN. The vertical dotted line depicts the onset of the WN period. (**F**) The distribution of jumps across the duration of the SCS from the UN group for each trial of CD2. Each dot represents a single jump event, and each tick on the x-axis represents the onset of each pip of tone or WN. The vertical dotted line depicts the onset of the WN period. (**G**) The percentage of the PA-R group that exhibited jumping during WN on CD2. Data are distributed across 1 s bins, each coinciding with one of the ten pips of WN that occurred during each SCS presentation. (**H**) The percentage of the UN-R group that exhibited jumping during WN on CD2. Data are distributed across 1 s bins, each coinciding with one of the ten pips of WN that occurred during each SCS presentation. (**I**) Total percentage of the cohort that jumped during WN over the whole CD2 session. (**J**) Total percentage of cohort that jumped to shock over the whole CD2 session.

The online version of this article includes the following source data for figure 3:

**Source data 1.** Dataset contains cumulative behavioral measurements for *Figure 3* from each group.

Similar analyses were performed for darting behavior. A high percentage of PA mice showed darting during WN (*Figure 4A*), darts were specific for the WN, and they were spread across the stimulus period (*Figure 4B*). Mice in the UN group almost never darted during the tone or WN (*Figure 4C and D*). Furthermore, PA mice displayed darts across the WN period on every trial (*Figure 4E*), whereas UN mice did not (*Figure 4F*). The PA group also had the largest percentage of mice that darted during WN compared to all other groups (*Figure 4G*). Interestingly, we did not detect darting from the reverse groups during the SCS in CD2. The non-PA groups only rarely expressed escape jumping or darting, yet they did have elevated activity indices (*Figure 2D*). Therefore, we measured the average distance traveled over the WN period, and we used a one-way ANOVA to determine if these higher activity levels were due to a simpler locomotor response. The groups did not differ in distance traveled during preSCS (*Figure 4H*, $F_{(3, 68)}=0.28$, p=0.84), but we did observe differences during tone (*Figure 4I*, $F_{(3, 68)}=3.34$, p=0.024) and WN (*Figure 4J*, $F_{(3, 68)}=9.83$, p<0.0001). During the tone, the UN group had a significantly greater distance traveled than the PA group (PA vs UN, p=0.014), which is reflective of the elevated freezing during the tone in the PA group (*Figure 2E*). During the WN, the PA group had greater distance traveled than the UN (PA vs UN, p=0.0018) and the PA-R groups (PA vs PA-R, p<0.0001), but not the UN-R group (PA vs UN-R, p=0.083). Finally, all groups darted to shock in similar percentages (*Figure 4K*).

In summary, associative pairings of SCS and shock produced significant cue-induced freezing to the tone, as well as robust jumping and darting behaviors that occurred over the entirety of WN presentations. Altering the contingency between WN and shock in the unpaired or reversed SCS conditions profoundly reduced these defensive behaviors, but the increased activity in all groups during WN suggests that inherent properties of the WN interact with non-associative processes to induce locomotor responses. This is reflected in the increased distance traveled during WN in all groups compared to the tone period. Representative behavioral responses of the PA, UN, PA-R, and UN-R groups to the SCS during conditioning are provided in *Video 1* and *Video 2*.

## Tone-evoked freezing in the PA group is reduced by extinction learning

We next analyzed how the defensive ethogram of each group changed over the course of two extinction sessions. A two-way ANOVA was used to analyze the effect of trial and group on freezing during the tone, and Tukey's multiple comparisons test was used for post-hoc comparisons. When analyzing tone-evoked freezing across extinction within the PA, UN, and SO groups (*Figure 5A*), a significant interaction between trial and group ($F_{(62, 2208)}=2.3$, p<0.0001) was found. For every trial except the last, the PA group exhibited a higher level of freezing during tone compared to the control groups (p<0.05 for Trials 1–15, for both sessions). Tone-evoked freezing presented similarly between the PA-R and UN-R groups (*Figure 5B*), yielding a significant interaction between trial and group ($F_{(31, 576)}=1.63$, p=0.018). The PA-R group maintained consistently high freezing during the tone across both extinction sessions, freezing more than the UN-R group for nearly all trials (p<0.05 for Trials 1, 3–16 on Ext1, all trials on Ext2).

To quantify the relative change in freezing over each extinction session, we calculated the difference in freezing between the first four trials and the last four trials for each session. An ordinary one-way ANOVA was used to analyze the effect of the group on changes in freezing during the tone, and Tukey's multiple comparisons test was used for post-hoc comparisons. There was a significant

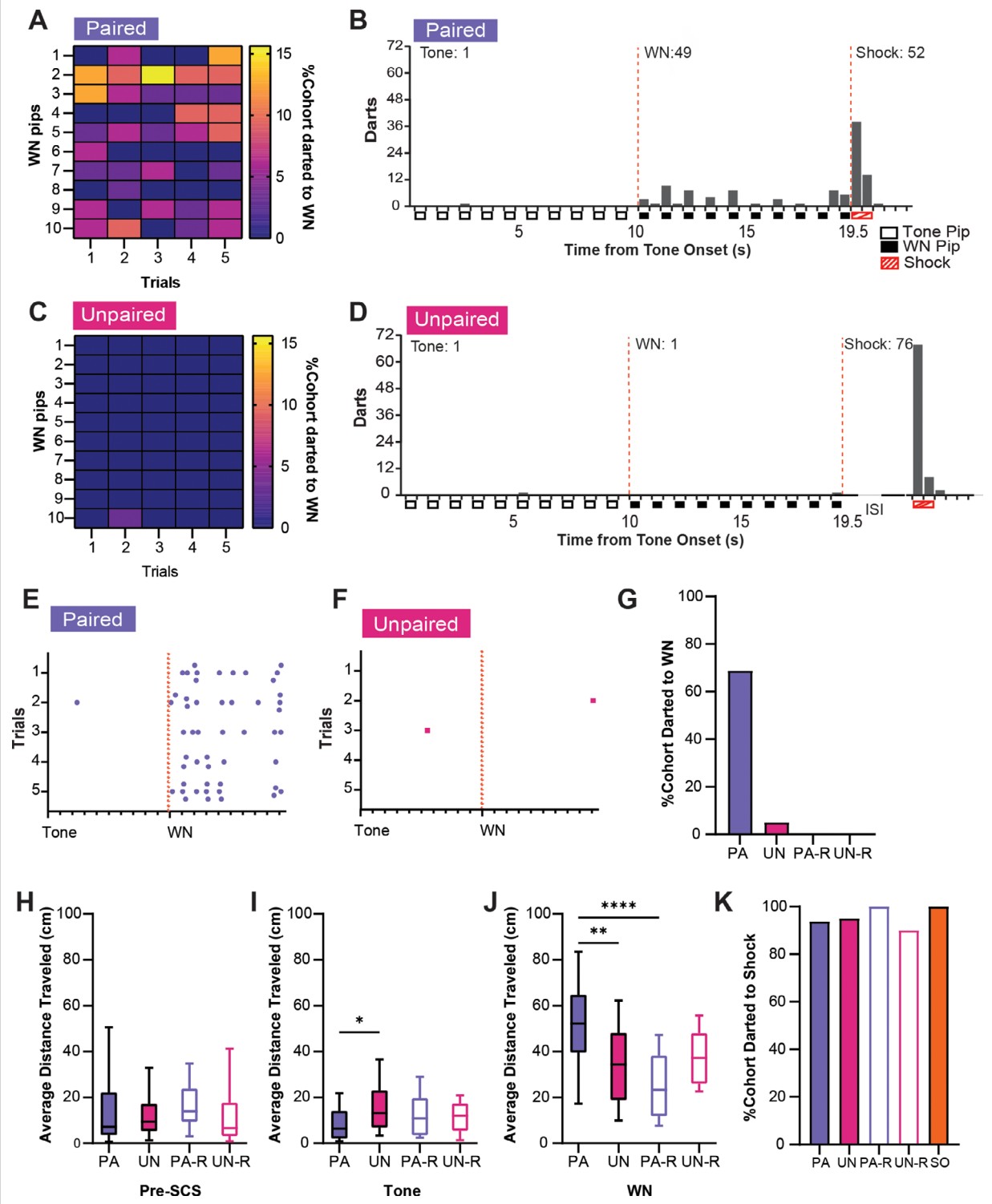

**Figure 4.** Associative serial compound stimulus (SCS)-shock pairings elicit darting responses to white noise (WN) during CD2. (**A**) The percentage of the paired (PA) group that exhibited darting responses to WN. Data are distributed across 1 s bins, each coinciding with one of the ten pips of WN that occurred during each SCS presentation. (**B**) The cumulative distribution of darts from 20 randomly selected subjects of the PA group across the duration of the SCS. Empty boxes represent each 0.5 s pip of tone, filled boxes represent each 0.5 s pip of WN, and striped boxes represent the 1 s shock stimulus. The vertical dotted lines depict the onset and termination of the WN period. Total darts per stimulus are listed above the histogram. (**C**) The percentage of the unpaired (UN) group that exhibited darting responses during WN. Data are distributed across 1 s bins, each coinciding with one of the ten pips of WN that occurred during each SCS presentation. (**D**) The cumulative distribution of darts from the UN group across the duration

*Figure 4 continued on next page*

*Figure 4 continued*

of SCS. Empty boxes represent each 0.5 s pip of tone, filled boxes represent each 0.5 s pip of WN, and striped boxes represent the 1 s shock stimulus. The vertical dotted lines depict the onset and termination of the WN period. ISI represents the period between SCS and shock. Total darts per stimulus are listed above the histogram. (**E**) The distribution of darts across the duration of SCS from 20 randomly selected subjects of the PA group. Each dot represents a single dart event, and each tick on the x-axis represents the onset of each pip of tone or WN. The vertical dotted lines depict the onset of the WN period. (**F**) The distribution of darts across the duration of SCS from the UN group. Each dot represents a single dart event, and each tick on the x-axis represents the onset of each pip of tone or WN. The vertical dotted lines depict the onset of the WN period. (**G**) The total percentage of each group that jumped during WN over the whole session. (**H**) Average distance traveled during the preSCS period. (**I**) Average distance traveled during the tone period. (**J**), Average distance traveled during the WN period. (**K**) The total percentage of each group that jumped to shock over the whole session. PA: n=32, UN: n=20, PA-R: n=10, UN-R: n=10, SO: n=20. Data from (**H-J**) are presented as box-and-whisker plots from min to max and were analyzed with one-way ANOVA and Tukey's post-hoc multiple comparisons test. *p<0.05; **p<0.01; ****p<0.0001.

The online version of this article includes the following source data for figure 4:

**Source data 1.** Dataset contains cumulative behavioral measurements for *Figure 4* from each group.

difference between groups on the first day of extinction (*Figure 5C*, $F_{(4, 87)}$=11.9, p<0.0001), with the PA-R group being the only one to increase freezing during tone over the session (PA-R vs PA, p<0.0001; PA-R vs UN, p<0.0001; PA-R vs UN-R, p=0.007; PA-R vs SO, p=0.0032). Freezing for the PA and UN groups decreased similarly over the session (PA vs UN, p=0.99), and the SO group had significantly less change in freezing compared to the PA group (PA vs SO, p=0.021), which was attributed to the low level of freezing during the tone in the SO group (*Figure 2A*). A significant difference between groups was also detected for the second extinction session (*Figure 5D*, $F_{(4, 87)}$=19.2, p<0.0001). Only the PA group exhibited a decrease in freezing during tone compared to all other groups (PA vs UN, p<0.0001; PA vs PA-R, p=0.0007; PA vs UN-R, p<0.0001; PA vs SO, p<0.0001).

To determine if freezing during the tone was cue-induced, or was simply a continuation of contextual freezing, we calculated the difference between freezing in the pre-SCS period and freezing during the tone for each extinction session (*Figure 5E and F*). An ordinary one-way ANOVA was used to analyze the effect of the group, and Tukey's multiple comparisons test was used for post-hoc comparisons. For both the first (*Figure 5E*, $F_{(4, 87)}$=105.5, p<0.0001) and second (*Figure 5F*, $F_{(4, 87)}$=137.5, p<0.0001) sessions of extinction, only the PA and PA-R groups increased freezing levels during the tone (p<0.0001 for all PA and PA-R comparisons to other groups for both sessions). The PA-R group showed a greater change in freezing from pre-SCS to tone than the PA group (Ext1, PA vs PA-R, p=0.0003; Ext2, PA vs PA-R, p<0.0001), whereas the UN, UN-R, and SO groups had equivalent freezing during the pre-SCS and tone periods (p>0.25 for all pairwise comparisons that excluded the PA and PA-R groups for both sessions). Taken together, these data suggest that the PA and PA-R groups associated the tone with threat, while freezing in the UN, UN-R, and SO groups was more indicative of contextual fear. Interestingly, while pairing either order of SCS with shock resulted in greater freezing during tone, we observed a phenotype more resistant to extinction within the PA-R group. These data indicate that the extinction of cue-induced freezing in the conditioned flight paradigm depends on its proximity to conditioned threat.

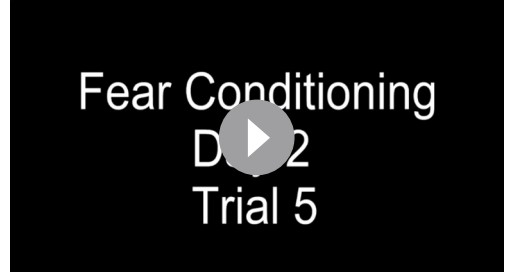

**Video 1.** Representative comparison of paired (PA) and unpaired (UN) groups' response to serial compound stimulus (SCS) during conditioning. The video features audio of the SCS, which consists of 10 pips of tone followed by 10 pips of white noise (WN).
https://elifesciences.org/articles/90414/figures#video1

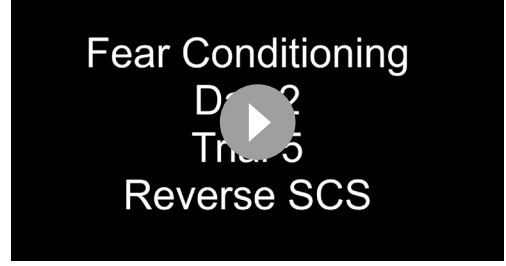

**Video 2.** Representative comparison of paired reverse (PA-R) and unpaired reverse (UN-R) groups' response to reverse serial compound stimulus (SCS) during conditioning. The video features audio of the reverse SCS, which consists of 10 pips of white noise (WN) followed by 10 pips of tone.
https://elifesciences.org/articles/90414/figures#video2

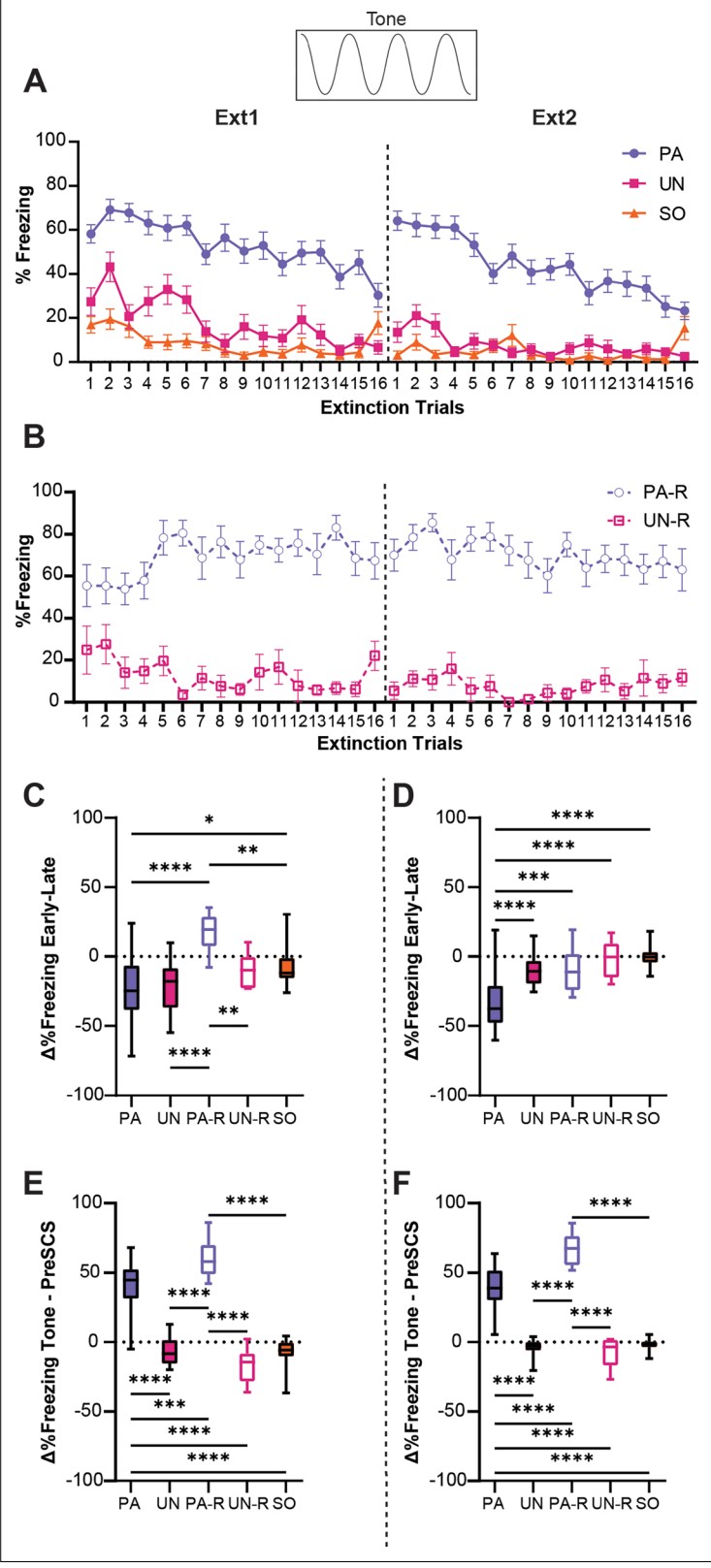

**Figure 5.** Tone-evoked freezing in the paired (PA) group is reduced by extinction learning. (**A**) Percent freezing during the tone period for the PA, unpaired (UN), and shock-only (SO) groups. (**B**) Percent freezing during the tone period for the paired reverse (PA-R) and unpaired reverse (UN-R). (**C**) The difference in average freezing during the tone period between the first and last four-trial bins of Ext1. (**D**) The difference in average freezing during the

*Figure 5 continued*

tone period between the first and last 4-trial bins of Ext2. (**E**) The difference in average freezing between pre-serial compound stimulus (SCS) and tone periods during Ext1. (**F**) The difference in average freezing between pre-SCS and tone periods during Ext2. PA: n=32, UN: n=20, PA-R: n=10, UN-R: n=10, SO: n=20. Data from (**A and B**) are presented as Mean ± SEM and were analyzed with two-way ANOVA and Tukey's post-hoc multiple comparisons test. Data from (**C–F**) are presented as box-and-whisker plots from min to max and were analyzed with one-way ANOVA and Tukey's post-hoc multiple comparisons test. *p<0.05, **p<0.01, ***p<0.001, ****p<0.0001.

The online version of this article includes the following source data for figure 5:

**Source data 1.** Dataset contains behavioral measurements for *Figure 5* from each individual mouse in each group, classified by sex, group, and trial where applicable.

## Stimulus-induced flight is associative and is partially replaced by freezing during extinction

Activity indices were calculated to analyze the effect of extinction learning on behavioral responses during the WN. The PA, UN, and SO groups had elevated activity indices in the early trials of extinction training, yet only the PA group showed a decrease in WN-evoked activity (*Figure 6A*). A two-way ANOVA was used to analyze the effect of trial and group, and Tukey's multiple comparisons test was used for post-hoc comparisons. There was a significant interaction between the trial and group ($F_{(62, 2208)}$=1.9, p<0.0001). Starting with the fifth trial of the first extinction session, the PA group expressed significantly less activity to WN compared to the UN and SO groups (p<0.05 compared to UN and SO for Trials 5, 6, and 8–13), and PA activity index scores remained below 1 for the entirety of the second extinction session (p<0.05 compared to UN and SO for Trials 1, 2, and 4–16). For the UN and SO groups, WN-evoked activity indices remained above 1 across extinction trials, indicating levels of movement that were higher during the WN than the pre-SCS period. There was no interaction between the trial and group (*Figure 6B*, $F_{(31, 576)}$=0.7, p=0.89) for the PA-R and UN-R groups. There was an effect of group ($F_{(1, 576)}$=64.7, p<0.0001), with activity being higher in the UN-R compared to the PA-R group. There was no significant effect of the trial ($F_{(31, 576)}$=0.66, p=0.92).

To illustrate the change in WN response over extinction, we calculated the difference in speed during WN between the first four and last four trials of each session. An ordinary one-way ANOVA was used to analyze the effect of the group, and Tukey's multiple comparisons test was used for post-hoc comparisons. The PA group showed a significant decrease compared to the UN-R and SO groups (*Figure 6C*, $F_{(4, 87)}$=6.4, p<0.0001; PA vs UN-R, p=0.0052; PA vs SO, p=0.0003) and a near-significant decrease to the UN group (PA vs UN, p=0.054). There were no significant differences between groups during the second extinction session (*Figure 6D*, $F_{(4, 87)}$=1.47, p=0.22).

Interestingly, as WN-evoked activity decreased during extinction, the PA group developed and maintained a freezing response to WN, while the other groups displayed almost no freezing to WN (*Figure 6E*). A significant interaction between trial and group was detected ($F_{(124, 2784)}$=1.3, p=0.019), and the PA group displayed greater freezing than the UN, UN-R, and SO groups for a majority of the first extinction session (p<0.05 for Trials 4, 6, and 10–16) and for most of the second session (p<0.05 for Trials 1–13). Additionally, the PA group froze more during WN than the PA-R group for several trials of the second session (p<0.05 for Trials 2–6, 8–10). When comparing changes in WN-evoked freezing between the first and last four trials for the first extinction session (*Figure 6F*, $F_{(4, 87)}$=4.2, p=0.0036), the PA group displayed significant increases compared to the UN and SO groups (PA vs UN, p=0.0086; PA vs SO, p=0.017), but did not differ from the PA-R or UN-R groups (PA vs PA-R, p=0.18; PA vs UN-R, p=0.22). For the second session (Fig 6G, $F_{(4, 87)}$=8.1, p<0.0001), the PA group exhibited a significant decrease in WN-evoked freezing compared to the UN, UN-R, and SO groups (PA vs UN, p=0.0005; PA vs UN-R, p=0.0006; PA vs SO, p=0.0015), but not the PA-R group (PA vs PA-R, p=0.85). Collectively, these findings show changes in the magnitudes and modes of behavior to WN within the PA group across extinction, indicating that WN-evoked flight in the PA group can be extinguished and is associative.

## Stimulus evoked escape jumping and darting during extinction

To determine if the activity measured during extinction was related to defensive flight or a more basic locomotor response, we examined the expression of jumping and darting behaviors between

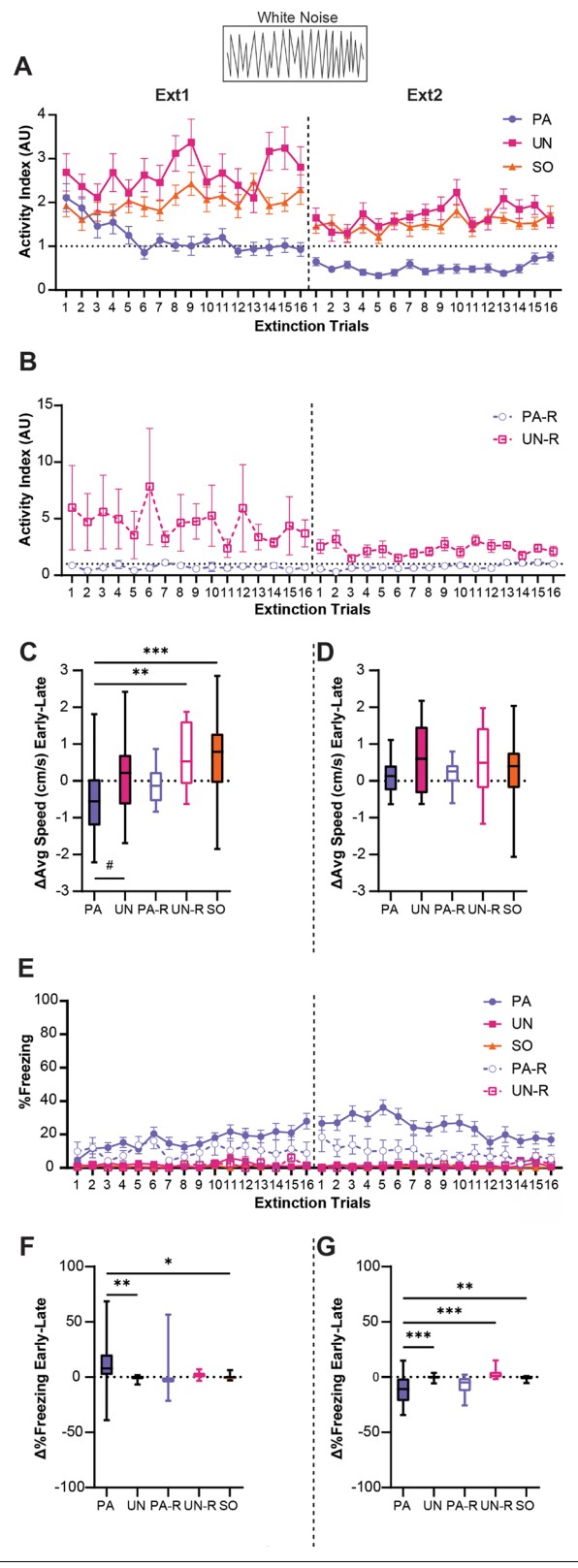

**Figure 6.** Stimulus-induced flight is associative and is partially replaced by freezing during extinction. (**A**) Trial-by-trial activity during the white noise (WN) period for the paired (PA), unpaired (UN), and shock-only (SO) groups during Ext1 and Ext2. (**B**) Trial-by-trial activity during the WN period for the paired reverse (PA-R) and unpaired reverse (UN-R) groups during Ext1 and Ext2. (**C**) Difference in average speed during the WN period from the first

*Figure 6 continued on next page*

*Figure 6 continued*

and last four-trial bins of Ext1. (**D**) Difference in average speed during the WN period from the first and last four-trial bins of Ext2. (**E**) Trial-by-trial freezing during the WN period for all groups during Ext1 and Ext2. (**F**) Difference in freezing during the WN period from the first and last four-trial bins of Ext1. (**G**) Difference in freezing during the WN period from early and late four-trial bins of Ext2. PA: n=32, UN: n=20, PA-R: n=10, UN-R: n=10, SO: n=20. Data from (**A, B and E**) are presented as Mean ± SEM and were analyzed with two-way ANOVA and Tukey's post-hoc multiple comparisons test. Data from (**C, D, F and G**) are presented as box-and-whisker plots from min to max and were analyzed with one-way ANOVA and Tukey's post-hoc multiple comparisons test. #p=0.054, *p<0.05, **p<0.01, ***p<0.001.

The online version of this article includes the following source data for figure 6:

**Source data 1.** Dataset contains behavioral measurements for *Figure 6* from each individual mouse in each group, classified by sex, group, and trial where applicable.

groups. Within the first four trials of the first extinction session, PA mice displayed the most jumping behavior during WN, with the SO group displaying only two jumps occurring near the onset of WN, and the UN group displaying no jumps (*Figure 7A*). When examining darting behavior within the first four trials of extinction, minimal darting was observed during the tone period, the PA and SO groups displayed darting behavior spread across the WN period, while the UN group darted only a few times (*Figure 7B*). Jumps and darts were not present within the second extinction session for any group.

During the first extinction session, only the PA group displayed a concentration of jumping responses during WN within the first block of trials (*Figure 7C*), and jumps to WN rarely occurred in any other group (*Figure 7E*). All groups displayed variable amounts of darting to WN, with the PA, PA-R, and SO groups having the largest proportions of darters within the session (*Figure 7D and F*). Notably, the increase in darting in the PA group occurs around Trial 7, which is approximately the time point at which jumping is fully extinguished. Within the PA-R and SO groups, WN-evoked darting is distributed throughout the session.

These data suggest that escape jumping is largely an associative response that switches to darting as the psychological distance of threat increases; however, darting is controlled by associative and non-associative mechanisms. Representative behavioral responses of the PA, UN, and SO groups to the SCS between early and late periods of the first extinction session are provided in *Video 3* and *Video 4*.

## Tail rattling is a non-associative behavioral response during extinction

We previously observed heightened tail-rattling responses during the early trials of fear conditioning, which decreased with further conditioning (*Borkar et al., 2020*). Given that tail rattling has been shown to increase in the presence of uncertain threat (*Salay et al., 2018*), we measured tail rattling in all groups during extinction to determine the effects of associative and non-associative mechanisms on this defensive response. During the first extinction session, tail rattling behavior during SCS presentations was more prevalent in the UN, UN-R, and SO control groups and was most prominent during the tone period (*Figure 8A and B*). To monitor tail rattling within groups, cumulative behavioral frequencies were taken from the first and last four trials within each extinction session (reverse SCS groups were excluded due to lower N-values). During early extinction, SO and UN mice displayed more tail rattling than PA mice during the tone (*Figure 8C*). The frequency of tail rattling was lower in all three groups during the second extinction session, yet the UN and SO groups both displayed more than the PA group (*Figure 8D*). All three groups displayed similar levels of tail rattling by the end of the second session. Over the first extinction session, a larger proportion of the UN and SO groups displayed tail rattle during to tone (*Figure 8E*), and the UN group had the greatest proportion of tail rattle to WN (*Figure 8F*).

These data suggest that tail rattling during the SCS is mostly a non-associative defensive response that is suppressed when the SCS predictably signals threat.

## Discussion

This study investigated the contributions of associative and non-associative processes to the expression of cue-induced defensive behaviors. The results signify that associative pairings and a proximal

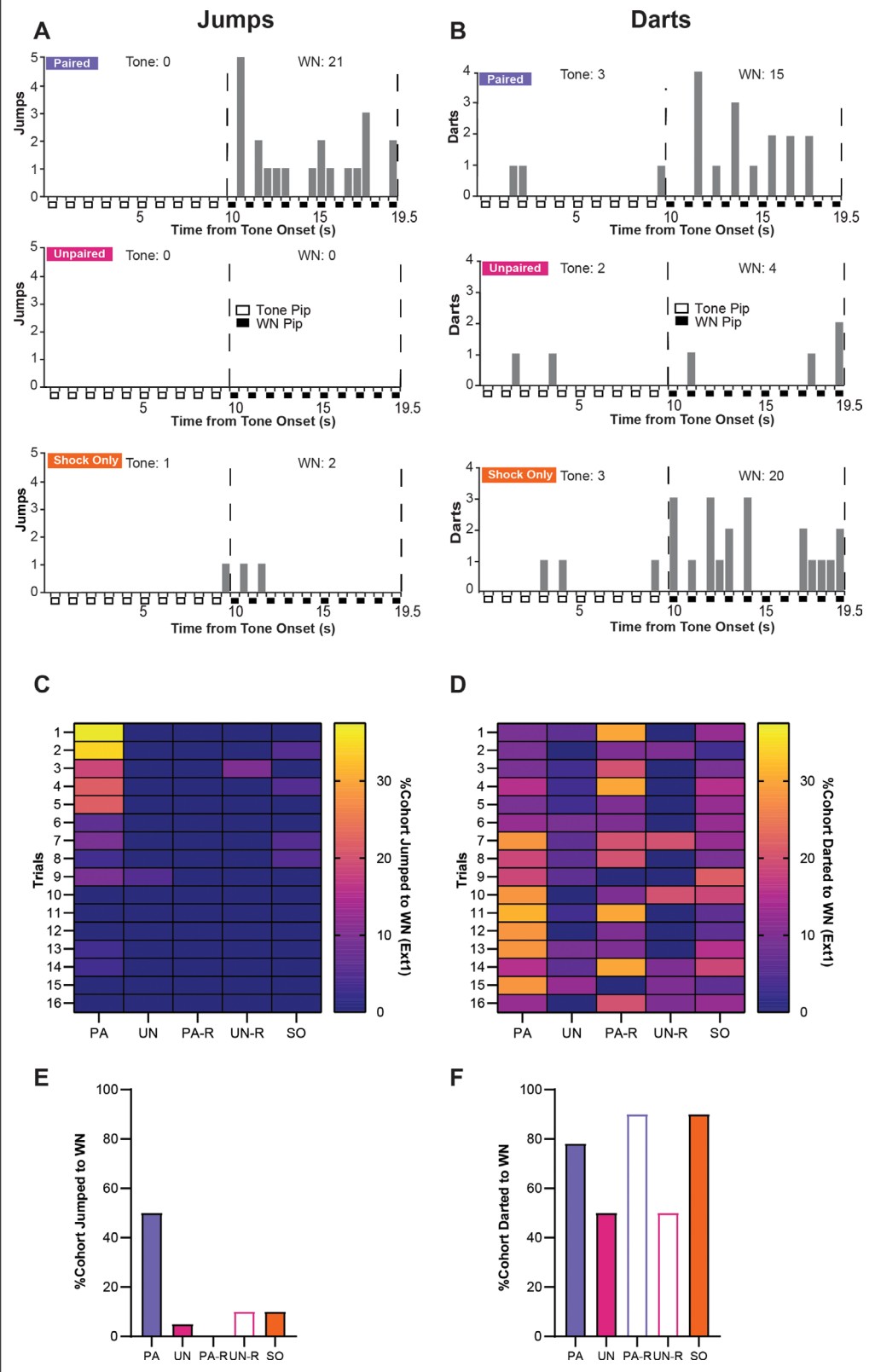

**Figure 7.** Stimulus-evoked escape jumping and darting during extinction. (**A**) The cumulative distribution of jumps from the first four trials of Ext1 for 20 randomly selected subjects from the paired (PA) group (*top*), the unpaired (UN) group (*middle*), and the shock-only (SO) group (*bottom*). Empty boxes represent each 0.5 s pip of tone, filled boxes represent each 0.5 s pip of white noise (WN), and the vertical dotted lines represent the onset

*Figure 7 continued on next page*

*Figure 7 continued*

and termination of the WN period. Total jumps per stimulus are listed above the histogram. (**B**) The cumulative distribution of darts from the first four trials of Ext1 for 20 randomly selected subjects from the PA group (*top*), the UN group (*middle*), and the SO group (*bottom*). Empty boxes represent each 0.5 s pip of tone, filled boxes represent each 0.5 s pip of WN, and the vertical dotted lines represent the onset and termination of the WN period. Total darts per stimulus are listed above the histogram. (**C**) The percentage of each group that exhibited jumping responses during the WN period of serial compound stimulus (SCS) per trial on Ext1. (**D**) The percentage of each group that exhibited darting responses to the WN period of SCS per trial on Ext1. (**E**) Total percentage of each cohort that jumped to WN over the whole Ext1 session. (**F**) Total percentage of each cohort that darted to WN over the whole Ext1 session.

The online version of this article includes the following source data for figure 7:

**Source data 1.** Dataset contains cumulative behavioral measurements for *Figure 7* from each group.

stimulus-threat association during fear conditioning produce maximal expression of cue-induced freezing and flight responses. Non-associative elements such as cue salience, change in stimulus intensity, and shock-induced sensitization elicit stress-associated behaviors like tail rattling and activity bursts, as hypothesized before (*Trott et al., 2022*), but the addition of the WN-threat associative pairing contributes significantly to eliciting high-intensity defensive responses like escape jumping. Therefore, these associative and non-associative factors combine to produce distinct behavioral transitions between freezing and flight.

During conditioning, we observed distinct ethograms for the PA and UN groups in response to the SCS. Freezing to tone and activity to WN were both significantly higher in the PA group compared to the UN group (*Figure 2E and K*), highlighting the impact of SCS-shock contingency on the magnitude of defensive responses. Notably, comparable results were previously reported in rats conditioned using similar parameters (*Totty et al., 2021*). Additionally, we found that the PA and PA-R groups showed significant increases in freezing from pre-SCS to tone, while in the UN and UN-R groups, freezing to tone was no greater than contextual freezing, suggesting that the paired groups placed associative value on the tone (*Figure 2J*). Lastly, the PA group displayed much more intense flight behavior during WN than the PA-R group (*Figure 2L*), indicating that the proximity of WN to the footshock threat affects defensive scaling.

Previous studies have factored jumping behaviors into normalized measures to gauge conditioned flight behavior (*Fadok et al., 2017*; *Hersman et al., 2020*; *Borkar et al., 2020*), but given the increased activity indices in the PA, UN, and UN-R groups, we examined if the escape jumping we observed was associative or non-associative (*Figure 3*). We found that the PA group exhibited consistent jumping responses to the WN stimulus during conditioning that were not reproduced in the UN, PA-R, or UN-R groups. Others have found that a salient stimulus is sufficient to induce conditioned jumping responses after multiple sessions (*Furuyama et al., 2023*). Given that the PA-R group did not display consistent jumping behavior to WN or

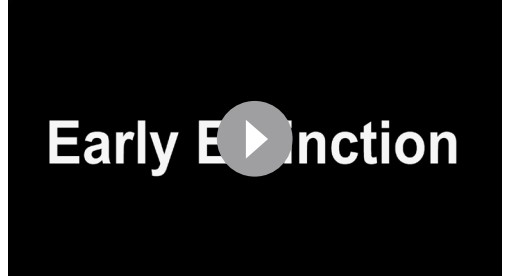

**Video 3.** Representative comparison of paired (PA), unpaired (UN), and shock-only (SO) groups' responses to the serial compound stimulus (SCS) during an early and late extinction trial of the first extinction session. The video features audio of the SCS, which consists of 10 pips of tone followed by 10 pips of white noise (WN).

https://elifesciences.org/articles/90414/figures#video3

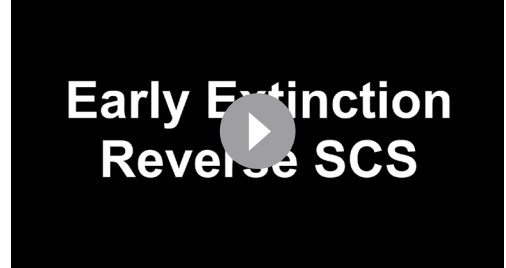

**Video 4.** Representative comparison of paired (PA)-R and unpaired (UN)-R groups' responses to the reverse serial compound stimulus (SCS) during an early and late extinction trial of the first extinction session. The video features audio of the reverse SCS, which consists of 10 pips of white noise (WN) followed by 10 pips of tone.

https://elifesciences.org/articles/90414/figures#video4

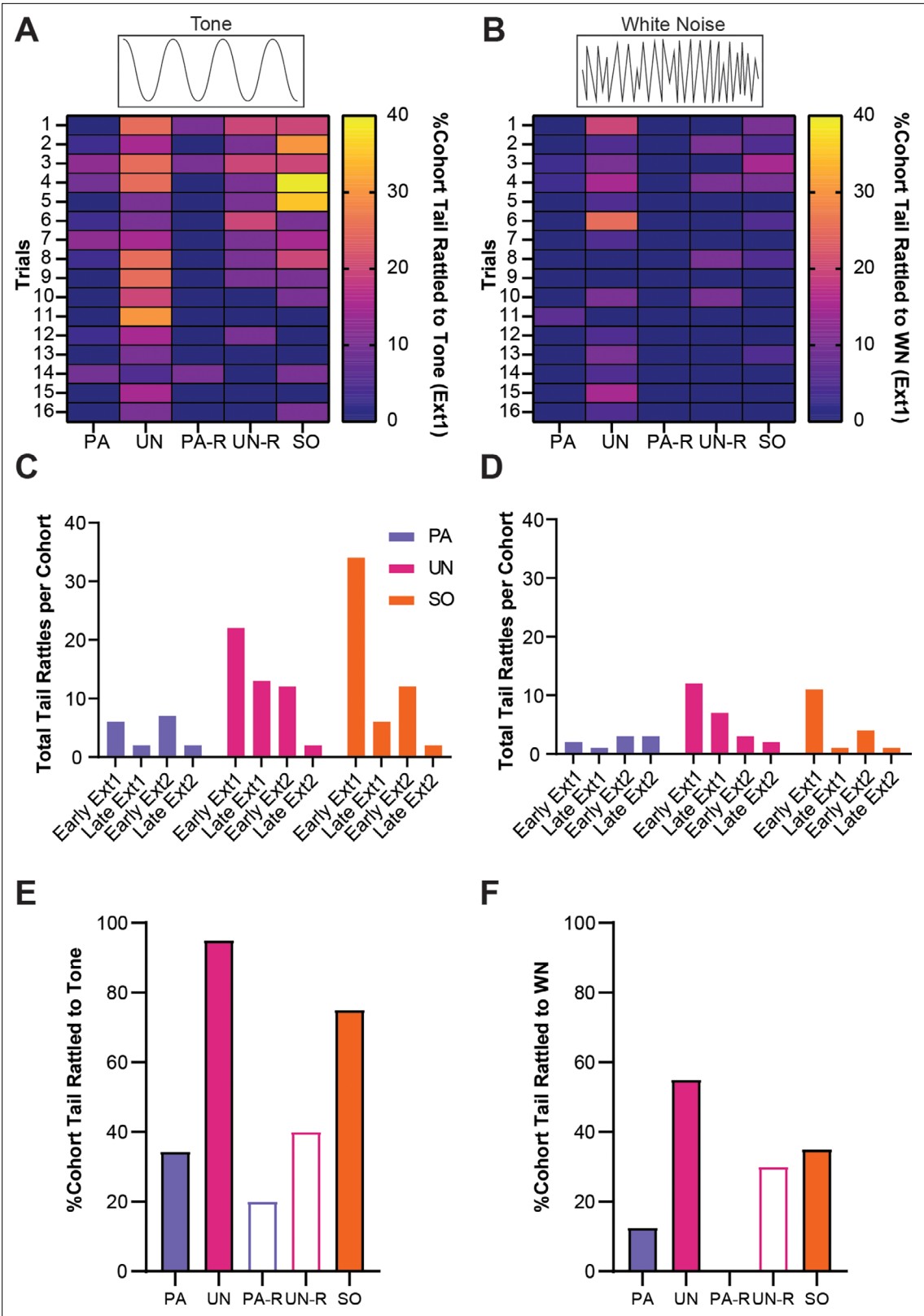

**Figure 8.** Tail rattling is a non-associative behavioral response during extinction. (**A**) The percentage of each group that exhibited tail rattling to the tone. (**B**) The percentage of each group that exhibited tail rattling to the white noise (WN). (**C**) Cumulative tail rattling during tone across early and late periods of Ext1 and Ext2. (**D**) Cumulative tail rattling during WN across early and late periods of Ext1 and Ext2. (**E**) Total percentage of each cohort that tail rattled to tone during Ext1. (**F**) Total percentage of each cohort that tail rattled to WN during Ext1.

*Figure 8 continued on next page*

*Figure 8 continued*

The online version of this article includes the following source data for figure 8:

**Source data 1.** Dataset contains cumulative behavioral measurements for *Figure 8* from each group.

tone, this suggests that a robust jumping response to WN in the SCS paradigm is not due to the non-associative salience of the WN as others have suggested (*Hersman et al., 2020*) but is instead due to its association with imminent threat.

Darting has been reported to be increased in rodents that undergo stress and fear conditioning, and it is more prevalent in female rats (*Gruene et al., 2015*; *Brzozowska et al., 2017*). Previous studies have measured darting to WN as a darts per minute measure (*Morena et al., 2021*; *Hoffman et al., 2022*; *Trott et al., 2022*) or as part of a composite escape score (*Hersman et al., 2020*), but our data suggests that examining darting requires a more detailed analysis. We found that the PA group exhibited darting across the entire WN period during conditioning, while the response was virtually nonexistent in the UN, PA-R, and UN-R groups (*Figure 4*). This is consistent with findings where conditioned darting has been shown to occur more often several seconds after a CS, rather than at its onset (*Mitchell et al., 2022*). This indicates that darting during conditioning is not caused by the salience of the tone-WN transition, but from an associative learned response. Additionally, the absence of darting in the PA-R group indicates that the temporal proximity of the WN to the shock also influences darting. Our data show that the control groups engage primarily in simple locomotor behaviors induced by stimulus salience (*Figure 4J*). Overall, darting behaviors during conditioning contribute to the higher activity score observed during WN in the PA group, and our data suggests that these contributions are the result of associative learning and threat imminence, rather than non-associative stimulus salience.

During extinction, the PA and PA-R groups showed the highest level of freezing to tone compared to the non-associative control groups (*Figure 5A and B*). This indicates that the paired groups had the strongest association between the tone stimulus and threat. This is further reinforced by the larger difference between tone-evoked freezing and pre-SCS freezing in the PA and PA-R groups (*Figure 5E and F*). Freezing during tone in the non-associative control groups, on the other hand, is no greater than freezing in the interstimulus intervals. Interestingly, tone-evoked freezing in the PA group under-went within-session extinction, while that of the PA-R group was resistant to extinction (*Figure 5A–D*). The sustained tone-evoked freezing over multiple extinction sessions in the PA-R group is likely a product of the temporal proximity of the cue to the footshock, which is set at a greater intensity (0.9 mA) in the conditioned flight paradigm than is traditionally used in Pavlovian threat conditioning (typically 0.2–0.6 mA). Because freezing during the tone is not elevated over contextual freezing levels in the unpaired and SO groups, the reductions in freezing during the tone during extinction in these groups can be attributed to reduced freezing overall.

In response to WN during extinction, the PA group transitioned from explosive circa-strike flight responses containing escape jumps to a combination of anticipatory post-encounter freezing and darting (*Figure 6A*, *Figure 7*). This likely reflects a larger perceived psychological distance from threat that influences defensive strategy (*Perusini and Fanselow, 2015*). The control groups showed only slight changes in WN-evoked activity over extinction, similar to responses of stressed mice to an unfa-miliar WN (*Figure 4*; *Hoffman et al., 2022*). The lack of decrease in WN-evoked activity over extinc-tion in the PA-R group compared to the PA group, and the lack of the control groups transitioning to freezing behavior, suggests that WN-evoked flight in the PA group is associative and is dependent on the perceived threat value of the WN stimulus.

An examination of jumping and darting behaviors over the first extinction session reveals a distinct ethogram in the PA group (*Figure 7*). The presence of jumping during early extinction trials in the PA group indicates that the WN initially signaled imminent threat, resulting in an explosive circa-strike escape response. However, the change from jumping to darting may reflect a change in perceived threat imminence in the same vein as the observed change from flight to freezing. This phenomenon was only observed in the PA group, indicating that this behavioral change is associative and is consis-tent with the predatory imminence continuum. This jumping behavior to WN is associative and can be extinguished, making it a suitable measure for future studies interested in how the nervous system controls experience-dependent high-intensity fear reactions.

The elevated activity indices from the UN and SO groups are similar to other studies that report heightened activity to WN after multiple footshocks, or a sudden change in stimulation (*Hoffman et al., 2022*; *Trott et al., 2022*). However, given the lack of darting and jumping from the UN group during extinction (*Figure 7A and B*), their increased activity is due to other locomotor behaviors unrelated to flight. The SO group maintained a consistent level of darting throughout the first extinction session, which contributed to their overall increased activity index (*Figure 7B and D*). Given that darting can be elicited in stressed mice (*Brzozowska et al., 2017*), it is probable that shock sensitization can prime an animal to dart more readily to an unfamiliar, but highly salient, stimulus upon stimulus transition. Indeed, darting behavior has been shown to change based on multiple parameters, decreasing both with increased shock intensity (*Mitchell et al., 2022*) and with prolonged extinction training (*Demars et al., 2022*), suggesting that darting is both an associative conditioned response to stimuli associated with threat and a response resulting from non-associative sensitization.

Within the UN, UN-R, and SO control groups, we observed a higher degree of tail rattling responses during extinction compared to the PA and PA-R groups, and tail rattling was more prominent during the tone period (*Figure 8*). Tail rattling has been observed in mice when determining hierarchical relationships (*Haber and Simmel, 1976*; *Terranova et al., 1998*; *Dorofeikova et al., 2023*), anticipating fighting (*Miczek et al., 2001*), and encountering looming threat (*Salay et al., 2018*). Thus, tail rattling may be a behavior elicited in stressed mice in the face of uncertain threat. Previously, we found that tail rattling occurs most often during early trials of fear conditioning, with a prominent decline later in conditioning (*Borkar et al., 2020*). This is consistent with our results from the UN, UN-R, and SO groups, who displayed greater tail rattling to unpaired (UN, UN-R groups) or novel (SO group) SCS presentations within the conditioning context, with reductions in tail rattling over the course of extinction exposure. Taken together, this suggests that tail rattling is not a behavior exhibited during post-encounter or circa-strike levels of threat, but rather within stressful scenarios where danger is uncertain but anticipated. Future studies that are interested in measuring defensive responses to threat signals such as context, odor, or innately aversive sensory stimuli should consider measuring tail rattling as a marker of non-associative anticipatory fear.

Current studies are investigating behaviors beyond freezing within classical Pavlovian conditioning paradigms (*Tryon et al., 2021*; *Laine et al., 2022*), but responses like jumping and darting are not always reliably elicited during CS presentation (*Colon et al., 2018*; *Akmese et al., 2023*; *Biddle and Knox, 2023*). Using the SCS flight conditioning paradigm, we elicit a robust continuum of consistent associative defensive responses during CS presentation that are seldom observed within classical Pavlovian conditioning. While behaviors like darting and tail ratting can occur due to non-associative stimulus-inherent properties, transitions between freezing and jumping are robustly present when associative factors signal threat imminence. Future studies can utilize this paradigm to investigate neuronal mechanisms that contribute to threat association and direct dynamic responses to threat, with important implications for developing new treatments for those who suffer from fear and anxiety disorders.

## Materials and methods

### Animal subjects

We used C57BL/6 J mice (Jackson Laboratory, Bar Harbor, Maine, Stock #000664), aged 3–6 mo in this study. Equal numbers of males and females were used in all experiments. All mice were individually housed on a 12 hr light/dark cycle throughout the study with ad libitum access to food and water. Behavioral experiments were performed during the light cycle. All animal procedures were performed in accordance with institutional guidelines and were approved by the Institutional Animal Care & Use Committee of Tulane University.

### Apparatus

Behavioral testing was performed in two contexts. Context A consisted of a 30 cm diameter transparent acrylic cylinder with a smooth acrylic floor, cleaned with 1% acetic acid between each subject. Context B consisted of a modular fear conditioning chamber (ENV-307W, Med Associates, Inc, Fairfax, Vermont) with metal grid flooring and walls of polycarbonate, stainless steel, and polyurethane, cleaned with 70% ethanol solution between sessions. Alternating current footshocks (ENV-414S, Med

Associates, Inc) were delivered to the mice during conditioning in Context B. A programmable audio generator (ANL-926, Med Associates, Inc) generated auditory stimuli that were delivered at 75 dB in each context via an overhead speaker (ENV-224AM, Med Associates, Inc). A serial compound stimulus (SCS) was used as previously described (*Fadok et al., 2017*; *Borkar and Fadok, 2021*; *Borkar et al., 2020*). The SCS consisted of ten pips of tone (7.5 kHz, 0.5 ms at 1 Hz) followed by ten pips of white noise (0.5ms at 1 Hz), and the reversed SCS consisted of ten pips of white noise followed by ten pips of tone. Behavioral protocols were generated using Med-PC software (Med Associates, Inc) to control auditory stimuli and shock with high temporal precision.

## Experimental design: SCS conditioning and extinction paradigm

Mice were randomly allocated to one of five groups: Paired (PA), Paired Reverse (PA-R), Unpaired (UN), Unpaired Reverse (UN-R), and Shock Only (SO). Behavioral testing took place over 5 d. For all days of the paradigm, PA-R and UN-R mice experienced the reversed SCS at identical presentation timings as their respective PA and UN counterparts. On Day 1 (Pre-Exposure), subjects were placed in Context A for a baseline period of 3 min, followed by four presentations of the SCS with a pseudorandom inter-stimulus interval (ISI) period of 90–100 s and a period of 40 s following the final SCS presentation, totaling 590 s per session. Day 2 and Day 3 (Conditioning) took place in Context B. On each conditioning day (CD1 and CD2), mice were subjected to one of three conditions after a 3- min baseline period. For all groups, each conditioning session lasted 820 s. PA mice (n=16 males, 16 females) and PA-R mice (n=5 males, 5 females) were presented with five pairings of the SCS co-terminating with a 1 s, 0.9 mA footshock, with pseudorandom ISI periods of 90–150 s and a period of 60 s following the final footshock of the session. UN mice (n=10 males, 10 females) and UN-R mice (n=5 males, 5 females) were presented with pseudorandom presentations of SCS and footshocks separate from one another with ISI periods of 40–60 s, with a period of 90 s following the final stimulus of the session. Stimuli were ordered such that the SCS could not reliably predict footshock. PA, PA-R, UN, and UN-R mice all received the same number of SCS and footshock presentations, only differing by SCS-footshock contingency. SO mice (n=10 males, 10 females) did not receive presentations of the SCS during conditioning and were given five footshocks with pseudorandom ISI periods of 120–160 s each session, with a period of 80 s following the final shock of the session. For all groups, stimulus timing and ISI differed between CD1 and CD2 to avoid predictable anticipation of stimuli before presentation. Days 4 and 5 (Extinction) took place in Context B, and each session consisted of 16 presentations of the SCS with pseudorandom ISI periods of 60–120 s, with a period of 50 s following the final SCS of the session. Each Extinction session (Ext1 and Ext2) lasted 1910s. Subjects were sacrificed after the conclusion of behavioral testing. Experiments were conducted in cohorts of 8–10 mice for a total of 10 replicates.

## Behavioral recording and analysis

All sessions were recorded to video using a camera (Pike, Allied Vision, Stadtroda, Germany) mounted above the behavioral contexts with stimulus events encoded to the same files using TTL pulses (Omniplex, Plexon, Dallas, Texas). Contour tracking (Cineplex, Plexon) was used to automatically detect freezing based on frame-by-frame changes in pixels. Freezing behavior was defined as a complete cessation of movement for at least 1 s, and results were confirmed with an observer blinded to the condition. By determining a calibration coefficient using the known size of the behavioral context and the camera's pixel dimensions, speed (cm/s) was extracted using the animal's center of gravity. An activity index was calculated for each animal using a ratio of its speed during either the tone or WN stimulus (CS) period and its average speed from the combined 10 s periods prior to each SCS presentation (pre-SCS) during the session; the number of jumps performed during that stimulus period was then added to this ratio ($Speed_{CS}/Speed_{avg\ pre\text{-}SCS}+Jumps$). Previously we calculated flight scores per trial using speed from each trial's pre-SCS period (*Fadok et al., 2017*; *Borkar et al., 2020*), but here we utilized an average from all pre-SCS periods in our calculations to avoid denominators that were close to or equal to 0, a complication noted by other groups (*Hersman et al., 2020*). Reflecting this change, we now refer to this measure of locomotor change as an 'activity index' instead of a 'flight score' as before. Escape jumps and tail-rattling behaviors were manually classified by an observer blinded to the condition. Jumps were defined as the period where the mouse had all four paws above the chamber floor. Tail rattling was defined as rapid

back-and-forth vibrations of the tip and midsection of the tail. Darting behavior was detected and classified using machine learning software as described below and was defined as rapid bursts of movement across the floor of the chamber. Distance traveled over pre-SCS, tone, and WN periods was calculated per mouse by plotting its average speed per 0.5 s intervals and integrating the area under the curve.

When performing behavioral analyses that reported cumulative frequencies per group, 20 random subjects from the PA group were used to match the population sizes of the UN and SO control groups. Due to the lower number of subjects in the PA-R and UN-R groups, they were excluded from frequency-based comparisons.

### Analysis of darting using machine learning

Darts were scored using the program Simple Behavior Analysis (*Nilsson et al., 2020*) to generate a machine learning algorithm capable of automatically detecting the occurrence of the behavior of interest. To generate this model, top-down footage (640×480 pixel resolution, 30 frames per second) of 16 male and female C57BL/6 J mice that underwent SCS fear conditioning in Context B was collected and analyzed in DeepLabCut (DLC) (*Mathis et al., 2018*) to assign 8 tracking points (Nose, L Ear, R Ear, Center of Mass, L Flank, R Flank, Tail Base, Tail Tip) to subjects. The DLC markerless tracking model was generated using manually assigned points from ~2000 frames trained using the ResNet50 Neural Network for 125,000 iterations. 2370 frames containing darting behavior were identified and added to the training set for SimBA. The darting start point was defined as the first frame in which the mouse began accelerating from a resting position, and the endpoint was defined as the last frame before the mouse returned to a full stop. Once the model was generated, all videos from all subjects were analyzed using a discrimination threshold of.37 and a minimum duration of 266 ms (eight frames).

### Statistical analysis

Sample sizes for each group were justified via power analysis ($\alpha$=0.05, power = 80%). Data were analyzed for statistical significance using Prism 9 (GraphPad Software Inc, San Diego, California). For all tests, the definition of statistical significance was $p < 0.05$. All data were checked for normal distribution using the Shapiro-Wilk normality test ($\alpha$=0.05). For pairwise comparisons between groups, an unpaired t-test with Welch's correction was used to assess behavioral differences since all relevant datasets had normal distributions. One-way analysis of variance (ANOVA) was used to assess behavioral differences between all conditioning groups. Two-way ANOVA was used to assess interactions of time point and conditioning variant between groups, as well as interactions of stimulus and conditioning variant within groups. When either ANOVA yielded significant interactions, Tukey's post-hoc multiple comparisons test was used to detect significant behavioral differences between groups.

## Acknowledgements

This study was supported by the National Institute of Mental Health of the National Institutes of Health under award number R01MH122561 to JPF. The content is solely the responsibility of the authors and does not necessarily represent the official views of the National Institutes of Health.

## Additional information

### Funding

| Funder | Grant reference number | Author |
|---|---|---|
| National Institute of Mental Health | R01MH122561 | Jonathan P Fadok |

The funders had no role in study design, data collection and interpretation, or the decision to submit the work for publication.

## Author contributions
Quan-Son Eric Le, Data curation, Formal analysis, Supervision, Validation, Investigation, Visualization, Methodology, Writing – original draft, Writing – review and editing; Daniel Hereford, Software, Formal analysis, Writing – original draft, Writing – review and editing; Chandrashekhar D Borkar, Writing – original draft, Writing – review and editing; Zach Aldaco, Julia Klar, Formal analysis; Alexis Resendez, Writing – review and editing; Jonathan P Fadok, Conceptualization, Resources, Software, Supervision, Funding acquisition, Validation, Visualization, Writing – original draft, Project administration, Writing – review and editing

## Author ORCIDs
Quan-Son Eric Le ![ORCID] https://orcid.org/0000-0003-1409-7790
Jonathan P Fadok ![ORCID] https://orcid.org/0000-0002-0608-5151

## Ethics
This study was performed in strict accordance with the recommendations in the Guide for the Care and Use of Laboratory Animals of the National Institutes of Health. All of the animals were handled according to approved institutional animal care and use committee (IACUC) protocol (#1941) of Tulane University.

Reviewer #1 (Public review): https://doi.org/10.7554/eLife.90414.3.sa1
Reviewer #2 (Public review): https://doi.org/10.7554/eLife.90414.3.sa2
Author response https://doi.org/10.7554/eLife.90414.3.sa3

---

# Additional files

## Supplementary files
• MDAR checklist

## Data availability
All source datasets for Figures 2–8 are provided in supporting Excel spreadsheet files.

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
