## [Editor Report · eLife Assessment]

This study is deemed to be an **important** work that carefully deconstructs multi-faceted conditioned fear behavior in mice. The well-controlled experiments provide **convincing** data that will be of interest to other researchers in the field.

---

## [Referee Report · Reviewer #1 (Public review)]

Summary

The main goal of the study was to tease apart the associative and non-associative elements of cued fear conditioning that could influence which defensive behaviors are expressed. To do this, the authors compared groups conditioned with paired, unpaired, or shock only procedures followed by extinction of the cue. The cue used in the study was not typical; serial presentation of a tone followed by a white noise (or reversed) was used in order to assess switches in behavior across the transition from tone to white noise. Many defensive behaviors beyond the typical freezing assessments were measured, and both male and female mice were included throughout. The authors found changes in behavioral transitions from freezing to flight during conditioning as the tone transitioned into white noise, and a switch in freezing during extinction such that it became high during the white noise as flight behavior decreased. Overall, this was an interesting analysis of transitions in defensive behaviors to a serially presented cue consisting of two auditory stimuli during conditioning and then extinction.

Strengths

The highlights in this study were the significant switches in freezing and escape-like behaviors as the cue transitioned between the two auditory stimuli during fear conditioning, and then adjustment of those behaviors across extinction.

These main findings were a result of thorough behavioral analyses with key control groups (reversed stimulus order, unpaired conditioning, and shock only groups), assessing freezing, jumping, darting and tail rattling to try to parse out associative versus non-associative features of the behavioral profiles.

Weaknesses

While the detailed analyses of defensive behaviors in mice in a situation of signaled imminent threat adds valuable knowledge to those studying fear conditioning, the caveat is that it is unclear how broadly applicable these findings truly will be. It makes sense that similar transitions in defensive behaviors will occur across organisms, but each organism and each psychiatric disorder will have unique profiles.

---

## [Referee Report · Reviewer #2 (Public review)]

Summary:

The authors examined several defensive responses elicited during Pavlovian conditioning using a serial compound stimulus (SCS) as the conditioned stimulus (CS) and a shock unconditioned stimulus (US) in male and female mice. The SCS consisted of a tone pips followed by white noise. Their design included conditions in which mice were exposed to the CS and US in a paired fashion, in an unpaired fashion, or only exposed to the shock US, as well as paired and unpaired conditions that reversed the order of the SCS. They compared freezing, jumping, darting, and tail rattling across all groups during conditioning and extinction. During conditioning, strong freezing responses to the tone pips followed by strong jumping and darting responses to the white noise were present in the paired group but less robust or not present in the unpaired or shock only groups. During extinction, tone-induced freezing diminished while the jumping was replaced by freezing and darting in the paired group. Together, these findings support the idea that associative pairings are necessary for conditioned defensive responses.

Strengths:

The study has strong control groups including a group that receives the same stimuli in an unpaired fashion and another control group that only receives the shock US and no CS to test the associative value of the SCS to the US. The authors examine a wide variety of defensive behaviors that emerge during conditioning and shift throughout extinction: in addition to the standard freezing response, jumping, darting, and tail rattling were also measured.

The revised version has greatly strengthened this study by including additional control groups (e.g., reversing the order of the compound stimuli in both paired and unpaired conditions).

---

## [Author Response]

The following is the authors’ response to the original reviews.

The reviewers found this manuscript to present convincing evidence for associative and non-associative behaviors elicited in male and female mice during a serial compound stimulus Pavlovian fear conditioning task. The work adds to ongoing efforts to identify multifaceted behaviors that reflect learning in classic paradigms and will be valuable to others in the field. The reviewers do note areas that would benefit from additional discussion and some minor gaps in data reporting that could be filled by additional analyses or experiments.

We thank the reviewers and the editors for their thoughtful and constructive critiques of our manuscript. We have updated our manuscript with data from additional experiments as suggested by the reviewers, and we have significantly edited the text and figures to reflect these additions. Our detailed, point-by-point responses are below.

**Reviewer #1 (Public Review):**
The main goal of the study was to tease apart the associative and non-associative elements of cued fear conditioning that could influence which defensive behaviors are expressed. To do this, the authors compared groups conditioned with paired, unpaired, or shock only procedures followed by extinction of the cue. The cue used in the study was not typical; serial presentation of a tone followed by a white noise was used in order to assess switches in behavior across the transition from tone to white noise. Many defensive behaviors beyond the typical freezing assessments were measured, and both male and female mice were included throughout. The authors found changes in behavioral transitions from freezing to flight during conditioning as the tone transitioned into white noise, and a switch in freezing during extinction such that it became high during the white noise as flight behavior decreased. Overall, this was an interesting analysis of transitions in defensive behaviors to a serially presented cue consisting of two auditory stimuli during conditioning and then extinction.

We thank the Reviewer for their supportive insight.

There are some concerns regarding the possibility that the white noise is more innately aversive than the tone, inducing more escape-like behaviors compared to a tone, especially since the shock only group also showed increased escape-like behaviors during the white noise versus tone. This issue would have been resolved by adding a control group where the order of the auditory stimuli was reversed (white noise->tone).

We appreciate this concern, and we have added two additional groups to address this possibility. We have conducted the same experimental paradigm with 2 reverse-SCS groups (WN—tone), one with paired (new PA-R group), and one with unpaired (new UN-R group), presentations to shock during conditioning. These experiments revealed that during conditioning day 2 in both reverse order groups, WN causes reductions in freezing and increases in locomotor activity (see revised Figure 2D), an effect that is stronger in the UN-R compared to the PA-R group. This locomotor effect is neither darting nor escape jumping in the PA-R group (revised Figure 3G, I; Figure 4G). In the UN-R group, WN induces more activity than the PA-R group (Figure 2D), including some jumping at WN onset (Figure 3H), but no darting (Figure 4G). It is worth noting that WN does not elicit defensive behavior before conditioning at the sound intensity we use (75dB; see Fadok et al. 2017, Borkar et al. 2020, Borkar et al. 2024). Together, these results suggest that WN is an inherently more salient stimulus than tone, and it can elicit defensive behaviors in shock-sensitized mice through non-associative mechanisms. Indeed, stimulus salience is a key factor in this paradigm for inducing activity (see Hersman et al. 2020).

While the more complete assessment of defensive behaviors beyond freezing is welcomed, the main conclusions in the discussion are overly focused on the paired group and the associative elements of conditioning, which would likely not be surprising to the field. If the goal, as indicated in the title, was to tease apart the associative and non-associative elements of conditioning and defensive behaviors, there needs to be a more emphasized discussion and explicit identification of the non-associative findings of their study, as this would be more impactful to the field.

We have rewritten the Discussion to provide a greater emphasis on the findings of the study that are more related to non-associative mechanisms. For example, we argue that cue-salience and changes in stimulus intensity can induce non-associative increases in locomotor behavior and tail rattling in shock-sensitized mice.

**Reviewer #2 (Public Review):**
Summary:The authors examined several defensive responses elicited during Pavlovian conditioning using a serial compound stimulus (SCS) as the conditioned stimulus (CS) and a shock unconditioned stimulus (US) in male and female mice. The SCS consisted of tone pips followed by white noise. Their design included 3 treatment groups that were either exposed to the CS and US in a paired fashion, in an unpaired fashion, or only exposed to the shock US. They compared freezing, jumping, darting, and tail rattling across all groups during conditioning and extinction. During conditioning, strong freezing responses to the tone pips followed by strong jumping and darting responses to the white noise were present in the paired group but less robust or not present in the unpaired or shock only groups. During extinction, tone-induced freezing diminished while the jumping was replaced by freezing and darting in the paired group. Together, these findings support the idea that associative pairings are necessary for conditioned defensive responses.Strengths:The study has strong control groups including a group that receives the same stimuli in an unpaired fashion and another control group that only receives the shock US and no CS to test the associative value of the SCS to the US. The authors examine a wide variety of defensive behaviors that emerge during conditioning and shift throughout extinction: in addition to the standard freezing response, jumping, darting, and tail rattling were also measured.

We thank the Reviewer for their supportive appraisal of this study’s strengths.

Weaknesses:This study could have greater impact and significance if additional conditions were added (e.g., using other stimuli of differing salience during the SCS), and determining the neural correlates or brain regions that are differentially recruited during different phases of the task across the different groups.

In the revised manuscript, we have conducted experiments with 2 reverse-SCS groups (WN—tone): one with paired (new PA-R group), and one with unpaired (new UN-R group), presentations to shock during conditioning. These experiments revealed that during conditioning day 2 in both reverse order groups, WN causes reductions in freezing and increases in locomotor activity (see revised Figure 2D), an effect that is stronger in the UN-R compared to the PA-R group. This locomotor effect is neither darting nor escape jumping in the PA-R group (revised Figure 3G, I; Figure 4G). In the UN-R group, WN induces more activity than the PA-R group (Figure 2D), including some jumping at WN onset (Figure 3H), but no darting (Figure 4G). Indeed, stimulus salience is a key factor in this paradigm for inducing activity (see Hersman et al. 2020). Together, these results suggest that WN is an inherently more salient stimulus than tone, and it can elicit defensive behaviors in shock-sensitized mice through non-associative mechanisms. It is worth noting that WN does not elicit defensive behavior before conditioning at the sound intensity we use (75dB; see Fadok et al. 2017, Borkar et al. 2020, Borkar et al. 2024).

We agree that determining the neuronal correlates and brain regions that are involved in defensive ethograms at various stages within this paradigm is of great importance, but we feel that those experiments are beyond the scope of the current study, which is focused on identifying behavioral differences based on associative and non-associative factors.

**Reviewer #1 (Recommendations For The Authors):**
In LINES 72-73, authors say they used a "truly random procedure" as one of their control groups. Then in LINES 113-116, they describe this group as "unpaired" where the "SCS could not reliably predict footshock". Combined, it is unclear if this group is random or unpaired. The "truly random procedure" is defined, by the cited Rescorla paper, as "the two events are programmed entirely randomly and independently in such a way that some "pairings" of CS and US may occur by chance alone". So, truly random would indicate that the shock may occur during the cue, while unpaired indicates the shock was explicitly unpaired from the cue. If the authors used a random procedure, the groups need to be labeled as random, not unpaired, and the # of cues that happened to coincide with footshock per animal needs to be reported somewhere. If the authors used an unpaired procedure (which appears to be the case based on 40-60s ITI between SCS and footshock being reported), it needs to be clearer and consistent throughout that it was explicitly unpaired, as well as removing the claim in LINE 72-73 that they used a "truly random procedure".

We did indeed use an explicitly unpaired procedure. We have adjusted the text and figures to better reflect this, and we removed any mentions of randomness with regards to the presentations of SCS and footshock.

Despite the lack of significant sex differences, it would still be helpful if data panels with individual data points (e.g. Fig 2E-J), were presented as identifiable by sex (e.g. closed vs open circles for males vs females).

The revised manuscript now compares four or five groups per figure, making data presentation complicated. Providing the individual data points in each panel reduces figure clarity, therefore, we feel it is best to present the data as box-and-whisker plots without them. However, the source data files for each figure are available to the reader and the data are clearly labeled to be identifiable by sex.

Is it not odd that all groups showed similar levels of contextual freezing during the 3min baseline? If shocks are unsignaled in the UN and SO groups, one would expect higher levels of contextual freezing compared to a paired group.

We are not certain why one would expect higher levels of contextual freezing in the UN and SO groups compared to the PA group at the beginning of conditioning day 2. Another study also looked at baseline freezing in a contextual fear group (which is the same as shock only in our study) and in an auditory cued fear conditioning group within the conditioning context, and their data show that freezing during the baseline period is equivalent between groups (Sachella et al., 2022).

During baseline on Extinction Day 1, it does seem that the unpaired and SO groups tend to have higher freezing levels compared to the paired groups. Author response image 1 shows baseline freezing during the first 3 minutes of extinction day 1. After two days of conditioning in the conditioned flight paradigm, contextual freezing either is, or trends to be significantly higher in the UN, UN-R, and SO groups than the PA and PA-R groups.

**Author response image 1. sa3fig1:** Baseline Freezing levels for all groups during the first extinction session. Baseline period is defined as the first 180 seconds of the session, before any auditory stimulus was presented. PA, Paired; UN, Unpaired; SO, Shock Only; PA-R, Paired Reverse; UN-R, Unpaired Reverse. *p<0.05, **p<0.01, ****p<0.0001.

Do the tone and WN elicit similar levels of defensive behaviors in a naïve mouse? Or have the authors tested WN followed by tone? Is there a potential issue that the WN may be innately aversive which is then amplified with training? i.e. does a tone preferentially induce freezing while WN induces active behaviors, regardless of which sensory stimulus is temporally closer to the shock? If the change in behavior is really due to the pairing and temporal proximity to shock, then there should be increased jumps, etc to the tone if trained with WN->tone.

WN can indeed be used as an aversive stimulus under certain conditions and at sufficiently high decibel levels. In the conditioned flight paradigm, WN is presented at 75dB, which is below the threshold for eliciting an acoustic startle response in a C57BL/6J mouse (Fadok et al. 2009). Also, during pre-exposure, when animals are naïve to the SCS, tone and WN stimuli do not elicit defensive behaviors (see Fadok et al. 2017, Borkar et al. 2020, 2024).

As suggested by the Reviewer, during revision we have included reverse-SCS paired (PA-R) and unpaired (UN-R) groups to test for the role of stimulus salience and stimulus order on defensive ethograms. During conditioning day 2, the PA-R group exhibited little freezing to the WN, with a slightly elevated activity index, and they exhibited robust freezing during tone (revised Figure 2A-H). The activity during the WN in the PA-R group was significantly lower than that of the PA group (Figure 2L). The PA-R group also did not respond to WN with escape jumps or darting (Figure 3I, 4G). The UN-R group displayed greater activity during the WN than the UN and PA-R groups, but less activity than the PA group (Figure 2D, H). The UN-R group did not dart but this group displayed some jumping at WN onset (Figure 3H), like what was observed in the UN group.

These data suggest that WN has inherent, salient properties that can induce some non-associative activity after the mouse has been sensitized by shock (see also Hersman et al. 2020 for more detailed analysis of stimulus salience in the conditioned flight paradigm). However, only in the PA group is robust flight behavior (comprised of high numbers of escape jumps and darting) observed. Therefore, both stimulus salience and temporal order are important for eliciting transitions from freezing to flight.

Fig 3G/4G are hard for me to understand. The figure legends say they're survival graphs but the y-axis labels "Latency to initial jump/dart (% of cohort)" confuses me. What is the purpose of these graphs? Perhaps they are not needed. Or consider presenting them similar to Fig 7C, D as those were more intuitive and faster for me to grasp.

We had intended these plots to show that a greater proportion of the paired group jumps and darts during WN compared to the unpaired group, and that the percentage of the cohort that jumps and darts increases across conditioning trials. Because these graphs were not clear, we have removed them, and we have replaced them with graphs comparing total cohort percentages that jumped (Figure 3I) or darted (Figure 4G) over the whole CD2 session.

For the extinction data, I did not see within group analyses for within or between session fear extinction to the tone. So, for the paired group, were the last 4 trials of Ext 1 significantly lower than the first 4 trials? If not, then they did not show within-session extinction. Also, for the paired group, were the last 4 trials of Ext 1 significantly different than the first 4 trials of Ext 2? This would test for long-term retention and spontaneous recovery.

In the original submission and in the revised manuscript, we calculated a delta change score for freezing during tone in the early versus late blocks of 4 trials, and then we statistically compared these differences across groups (Figure 5C, D). This allowed us to assess between-group differences in changes to tone-evoked freezing during extinction. Freezing to tone did decrease significantly over the first extinction session for the paired group (Early Ext1 vs Late Ext1, paired t-test, t(31) = 6.23, p<0.0001), and when comparing late Ext1 and early Ext2, we found that tone-evoked freezing did significantly increase (Late Ext1 vs Early Ext2, paired t-test, t(31) = 5.26, p<0.0001). This increase in cue-induced freezing between days of extinction is characteristic of C57BL/6J mice (Hefner et al., 2008). Our study did not test for more distal timepoints, so we cannot comment on the efficacy of long-term retention or spontaneous recovery.

For the conditioning and extinction data across Figs 2, 5 and 6, what I gather from them is that freezing is high to the tone and low to the WN during conditioning, and then low to the tone, and high to the WN across extinction. Then for activity levels I see they are low to the tone and high to the WN during conditioning, and then low to the WN during extinction. The piece that is missing is what are activity levels like to the tone during extinction. Are they low like in conditioning and remain low in extinction? Or do they increase across extinction as freezing decreases? As I was going through these graphs I drew myself out step function summaries of the freezing and activity levels between tone/WN for conditioning vs extinction; maybe the authors could consider a summary figure.

We thank the Reviewer for their interest. We found that within the paired group, activity to tone remained low throughout both days of extinction (though increased within each session) and did not return to normal activity levels. We present this data in Author response image 2. We thank the Reviewer for the suggestion of a summary figure, but we feel there are too many axes of classification (between-group, within-group, multiple behaviors, tone/WN, conditioning/extinction) to coherently present our findings in a single figure.

**Author response image 2. sa3fig2:** Trial-by-trial plot of activity index during the tone period of SCS across both extinction sessions for the PA group. SCS, Serial compound stimulus; Ext, extinction; PA, Paired.

In the discussion (LINE 592-3), they discuss that shock sensitization in the SO group may prime a stressed animal to dart more readily to WN upon stimulus transition. Should this not also happen during the transition of silence to tone? What is special about a transition between two auditory stimuli that would result in panic like behavior in an animal that only received shock presentations? This also gets back to an earlier concern above regarding the potentially innately aversiveness of the WN.

After 2 days of shock sensitization, we observe that mice exhibit freezing to the tone during the first three trials of extinction day 1 (Figure 5A). This non-associative freezing response is like that observed in other studies of non-associative fear processing (please see Kamprath and Wotjak, 2004). As trials progress during extinction day 1, mice do become mildly activated during the tone (Author response image 3). The transition to WN in the shock-only group during extinction induces non-associative darting responses, but it does not induce escape jumping behavior (Figure 7). We hypothesize that the innate salience of the WN is a vital factor contributing to these escalated responses. The importance of stimulus salience in conditioned flight was also demonstrated by Hersman et al., 2020 for SCS conditioning, and by Furuyama et al., 2023 for single tone conditioning. Just as with conditional freezing responses (Kamprath and Wotjak, 2004), we believe that conditional flight is controlled by summative components, one being associative and the other non-associative.

**Author response image 3. sa3fig3:** Trial-by-trial plot of activity index during the tone period of SCS across both extinction sessions for the SO group. SCS, Serial compound stimulus; Ext, extinction; SO, Shock Only.

In the discussion (LINE 583), they say that the development of explosive defensive behaviors are "not achievable with traditional single-cue Pavlovian conditioning paradigms". The authors should include a caveat here that the current study did not compare their results to a group of mice that received just WN-shock pairings.

We thank the reviewer for this comment. This statement was meant to highlight that traditional paradigms do not offer an element of signaling the temporal imminence of threat, only its inevitability. It was not our intention to state that defensive escape behaviors were unachievable in single-cue conditioning paradigms, and we regret not making this clear. Indeed, the supplement of Fadok et al. 2017 shows that WN-shock conditioning is capable of inducing flight, Furuyama et al. 2023 shows that tone-shock conditioning is capable of inducing flight under specific parameters, and Gruene et al. 2015 demonstrates that single CS-US pairings induce conditional darting behaviors in female rats. We have adjusted the text to better reflect our intent.

Minor comment to LINE 613-5: Speaking as someone who has done fear conditioning in both mice and rats, tail rattling may be specific to mice (I have seen this often) and likely not observable in rats (never seen it).

We thank the Reviewer for this information. We have adjusted our text to mainly discuss mouse-specific tail rattling.

**Reviewer #2 (Recommendations For The Authors):**
The research questions in this study are novel and bring new insight to the field. However, there are some issues that can be addressed to improve the overall quality of the study, namely, the reader is left wanting to know more, especially about how neural circuits contribute to these different defensive behaviors during this task. Below are some recommendations for the authors that would greatly improve the impact and significance of this study.(1) What are the neural correlates or circuits recruited during these different defensive behaviors across the course of conditioning and extinction? How might they differ between the PA and UN groups? What differences might emerge when an animal is shifting their defensive behavior from freezing to darting, for example? Answering these questions would require intensive additional experiments, therefore more discussion of possible neural mechanisms that might be recruited during this task would be appreciated, given the scope of the subject area.

We agree that understanding the neural circuits recruited during these behaviors and across conditioning and extinction is of vital importance. We are actively working on these questions, and we have published on the role of central amygdala circuits (Fadok et al. 2017) as well as on top-down control of flight by the medial prefrontal cortex (Borkar et al. 2024). Because the current manuscript is focused on learning mechanisms influencing defensive behavior, we would prefer to focus our discussion on that, rather than speculating on possible neural mechanisms. However, we have added a statement in the Discussion (LINES 706-707) emphasizing that future studies should investigate the neuronal mechanisms contributing to threat associations and different defensive behaviors.

(2) Were any vocalizations observed during conditioning or extinction phases? If not, could you speculate how type and occurrence of vocalizations might correlate with the different defensive responses observed?

Audible vocalizations were only observed during footshock presentations (squeaks). Unfortunately, we do not have the proper specialized recording equipment to monitor the full spectrum of mouse vocalizations, especially those in the ultrasonic range. Thus, we cannot speculate on the nuances of vocalizations in mice with respect to this behavioral paradigm. To the best of our knowledge, mice have not been reported to emit specific ultrasonic calls during conditioned threat like those of rats. That said, it would be of interest to determine if mice emit different vocalizations during different defensive behaviors.

(3) The transition from freezing to flight during the SCS is thought to be due to the close proximity of threat imminence between the WN CS and shock US. What if you switched the order of the SCS stimuli to WN followed by tone stimuli? If the salience of the WN stimulus is truly driving the jumping behavior, then it would be observed even if the WN stimulus preceded the pure tone stimulus and that would bring additional evidence that it is the associative value of the stimuli rather than its salience that's driving the defensive behaviors. What do you predict you would observe in rodents that were given a WN-tone SCS paired and unpaired in the same design of this study?

As suggested by the reviewer, we collected data from reverse-SCS paired and unpaired groups and reported our findings within the manuscript. Our detailed findings are also discussed above. Overall, we find that a combination of stimulus salience and temporal proximity, and a summation of non-associative and associative mechanisms, are necessary to elicit explosive flight behavior (escape jumping and darting).

References

Borkar CD, Dorofeikova M, Le QE, Vutukuri R, Vo C, Hereford D, Resendez A, Basavanhalli S, Sifnugel N, Fadok JP (2020) Sex differences in behavioral responses during a conditioned flight paradigm. Behavioural Brain Research 389:112623.

Borkar CD, Stelly CE, Fu X, Dorofeikova M, Le QE, Vutukuri R, Vo C, Walker A, Basavanhalli S, Duong A, Bean E, Resendez A, Parker JG, Tasker JG, Fadok JP (2024) Top-down control of flight by a non-canonical cortico-amygdala pathway. Nature 625: 743-749.

Fadok JP, Krabbe S, Markovic M, Courtin J, Xu C, Massi L, Botta P, Bylund K, Müller C, Kovacevic A, Tovote P, Lüthi A (2017) A competitive inhibitory circuit for selection of active and passive fear response. Nature 542:96-100.

Furuyama T, Imayoshi A, Iyobe T, Ono M, Ishikawa T, Ozaki N, Kato N, Yamamoto R (2023) Multiple factors contribute to flight behaviors during fear conditioning. Scientific Reports 13:10402.

Gruene TM, Flick K, Stefano A, Shea SD, Shansky RM (2015) Sexually divergent expression of active and passive conditioned fear responses in rats. eLIfe 4:e11352.

Hefner K, Whittle N, Juhasz J, Norcross M, Karlsson RM, Saksida LM, Bussey TJ, Singewald N, Holmes A (2008) Impaired Fear Extinction Learning and Cortico-Amygdala Circuit Abnormalities in a Common Genetic Mouse Strain. Journal of Neuroscience 6:8074-8085.

Hersman S, Allen D, Hashimoto M, Brito SI, Anthony T (2020) Stimulus salience determines defensive behaviors elicited by aversively conditioned serial compound auditory stimuli. elife 9:e53803.

Kamprath K and Wotjak CT (2004) Nonassociative learning processes determine expression and extinction of conditioned fear in mice. Learning & Memory 11:770-786.

Sachella TE, Ihidoype MR, Proulx CD, Pafundo DE, Medina JH, Mendez P & Piriz J (2022) A novel role for the lateral habenula in fear learning. Neuropsychopharmacology 47:1210-1219.